# Future shift in winter streamflow modulated by internal variability of climate in southern Ontario

Olivier Champagne[1]*, M. Altaf Arain[1], Martin Leduc[2], Paulin Coulibaly[1,3], Shawn McKenzie[1]

1 School of Geography and Earth Sciences and McMaster Centre for Climate Change, McMaster University, Hamilton, Ontario, Canada
2 Ouranos and Centre ESCER, Université du Québec á Montréal, Montréal, Québec, Canada
3 Department of Civil Engineering, McMaster University, Hamilton, Ontario, Canada

*Corresponding Author*: Olivier Champagne, Burke Science Building, Room 313, McMaster University, 1280 Main Street West, Hamilton, Ontario, L8S 4K1, Canada. Email: champago@mcmaster.ca. Tel: (905) 525-9140 ext. 27879

**Abstract.** Fluvial systems in southern Ontario are regularly affected by widespread early-spring flood events primarily caused by rain-on-snow events. Recent studies have shown an increase in winter floods in this region due to increasing winter temperature and precipitation. Streamflow simulations are associated with uncertainties mainly due to the different scenarios of Greenhouse gases emissions, general circulation models (GCM) or the choice of the hydrological model. The internal variability of climate, defined as the chaotic variability of atmospheric circulation due to natural internal processes within the climate system, is also a source of uncertainties to consider. Internal variability uncertainties can be assessed using hydrological models fed by downscaled Global Climate Model large ensemble (GCM-LE) data, but GCM output have a too coarse scale to be used in hydrological modelling. The Canadian Regional Climate Model large ensemble (CRCM5-LE), a 50-member ensemble downscaled from the Canadian global climate model version 2 large ensemble (CanESM2-LE), was developed to simulate local climate variability over northeastern North America under different future climate scenarios. In this study, CRCM5-LE temperature and precipitation projections under RCP 8.5 scenario were used as input in the Precipitation Runoff Modelling System (PRMS) to simulate streamflow at a near future horizon (2026-2055) for four watersheds in southern Ontario. To investigate the role of internal variability of climate in the modulation of streamflow, the 50-members were first grouped in classes of similar projected change in January-February streamflow and temperature-precipitation between 1961-1990 and 2026-2055. Then, the regional change in Geopotential height (Z500) from CanESM2-LE was calculated for each class. Model simulations showed an average January-February increase in streamflow of 18% ($\pm$8.7) in Big Creek, 30.5% ($\pm$10.8) in Grand River, 29.8% ($\pm$10.4) in Thames River and 31.2% ($\pm$13.3) in Credit River. 14% of all ensemble members projected positive Z500 anomalies in North America's East Coast enhancing rain, snowmelt and streamflow volume in January-February. For these members the increase of streamflow is expected to be as high as 31.6% ($\pm$8.1) in Big Creek, 48.3% ($\pm$11.1) in Grand River, 47% ($\pm$9.6) in Thames River and 53.7% ($\pm$15) in Credit River. Conversely, 14% of the ensemble projected negative Z500 anomalies in North America's East Coast and were associated with a much lower increase in streamflow: 8.3% ($\pm$7.8) in Big Creek, 18.8% ($\pm$5.8) in Grand River, 17.8% ($\pm$6.4) in Thames River and 18.6% ($\pm$6.5) in Credit River. These results provide important information to researchers, managers, policy makers

and society about the expected ranges of increase in winter streamflow in a highly populated region of Canada, and will help to understand how internal variability of climate is expected to modulate the future streamflow in this region.

## 1 Introduction

Increasing atmospheric greenhouse gases (GHG) concentration is projected to increase air temperature globally and modify the regional precipitation regimes (Hoegh-Guldberg et al., 2018). GHG-driven climate change is projected to impact watershed fluvial hydrological regimes especially in snow dominated regions (Barnett et al., 2005) with serious implications for flood management and water resources (Hamlet and Lettenmaier, 2007; Wu et al., 2015).

The quantification of streamflow and other hydrological processes using hydrological models is becoming an active area of research in various regions of the world. However, the use of hydrological models to project the future hydrology is subject to uncertainties (Clark et al., 2016) that have recently been intensely investigated (Leng et al., 2016). Part of the uncertainties are associated with the projections of climate through the choice of the Global Climate Model (GCM), the GHG emission scenario (Kour et al., 2016; Stephens et al., 2010) and the climate data downscaling method (Fowler et al., 2007; Schoof, 2013). In addition, temporal evolution of temperature and precipitation, simulated by the GCMs, is modulated by the internal variability of climate due to inherently chaotic internal processes within the climate system (Deser et al., 2014; Lorenz, 1963). These uncertainties are cascading to the hydrological processes and streamflow (Lafaysse et al., 2014) and additional uncertainties are associated to the choice of the hydrological models (Boorman et al., 2007; Devia et al., 2015) and model calibration techniques (Khakbaz et al., 2012; Moriasi et al., 2007).

The uncertainties due to the internal climate variability is one of the biggest sources of uncertainty for the early 21st century hydrological projections (Harding et al., 2012; Hawkins and Sutton, 2009; Lafaysse et al., 2014). The internal variability of climate is a cause of the hiatus observed in global warming in the 2000s (Dai et al., 2015) and is expected to mask the impact of human-induced climate change on precipitation (Rowell, 2012) and streamflow (Zhuan et al., 2018). To assess the contribution of internal variability of climate in the overall climate-change projections uncertainty, GCM large ensembles (GCM-LE), based on small initial condition variations between members of the ensemble, have been used recently (Deser et al., 2014; Kay et al., 2015; Kumar et al., 2015). This method was used to investigate how these uncertainties are transferred to hydrological processes in large watersheds (Gelfan et al., 2015). However, such coarse scale GCMs data should be downscaled to be used in small watersheds (Fowler et al., 2007). Despite the fact that Regional climate models are a computationally costly downscaling method (Lafaysse et al., 2014; Thompson et al., 2015), Regional Climate Model large ensembles (RCM-LEs) offer the possibility to relate each member of a RCM to large scale variability from GCM-LEs. Furthermore, RCM-LEs avoid additional and ambiguous sources of uncertainty caused by the statistical methods (Gelfan et al., 2015). One such dataset is the Canadian Regional Climate Model large ensemble (CRCM5-LE), a 50-members high resolution (12 km grids) regional model ensemble dataset produced over northeastern North America, that has recently been

developed as part of the Québec-Bavaria international collaboration on climate change project (ClimEx project; Leduc et al., 2019).

In the literature, several studies have projected an increase in winter streamflow in the Great Lakes region due to earlier snowmelt and increase in precipitation (Byun et al., 2019; Erler et al., 2018; Grillakis et al., 2011; Kuo et al., 2017) but the role of internal variability of climate was the subject of very few studies. Large ensembles have been previously used as input in multiple hydrological models in the Au Saumon catchment in southern Québec (Seiller and Anctil,2014) and in the Grand River watershed in southern Ontario (Erler et al., 2018). However, theses studies only used a few ensemble members which remove the possibility of assessing a large range of internal variability in the projections of the future hydrological responses.

The main goal of this study is to explore the impact of internal variability of climate in the projections of hydrologic processes and winter streamflow in major watersheds in southern Ontario in the Great Lakes region. Great Lakes region contains ~20 per cent of the world's unfrozen surface freshwater, while southern Ontario is home to one third of Canadian population (Statistics Canada, 2016). The specific objectives of this study are to (i)  Project the future evolution of streamflow in four watersheds in southern Ontario, using the Precipitation Runoff Modelling System (PRMS) forced by a large 50-members ensemble (CRCM5-LE) under IPCC RCP 8.5 scenario and (ii) Investigate the impact of the future projected changes in the regional atmospheric circulation on the hydrologic processes and winter streamflow in these watersheds.

## 2 Methods

### 2.1 Study area

Southern Ontario is a humid region according to the Köppen Geiger climate classification (Kottek et al., 2006), with an average annual precipitation of 1000 mm. The precipitation is well distributed throughout the year and about 200 mm falls as snow in the winter (Wang et al., 2015). The amount of rain and snow varies spatially due to the presence of Great Lakes. In winter the amount of snow is enhanced close to Lake Huron and Georgian Bay by lake effects (Suriano and Leathers, 2017) while in summer the precipitation are lower near the lakes because the convection is inhibited (Scott and Huff, 1996). The region is characterized by a mixed flood regime with high flows generated by rain, snowmelt and rain on snow events occurring from late February to early April (Burn and Whitfield, 2015). These events are occurring earlier recently due to a higher contribution of rainfall to the overall winter precipitation (Burn and Whitfield, 2015).

Four watersheds (Big Creek, Credit River, Grand River and Thames River) were selected for this study considering their long hydrometric data and representation of the diversity of spatial scales, soil type, and land use in this region (Figure 1 and Table 1). Agriculture activity is the largest land use category in all four watersheds, covering more than 80% of the entire surface in Big Creek, Thames River and Grand River. Credit River has the highest proportion of forest (32%), mostly deciduous species. Several major urban areas are located in the study area: Brantford, Cambridge, Kitchener-Waterloo and

Guelph in the Grand River watershed; and London in the Thames River. Additional urban areas are located in the Credit river watershed in the vicinity of the Greater Toronto Area while the Big Creek watershed contains the lowest proportion of urbanization (2%). These watersheds also vary in soil type: sand predominates in Big Creek (79%) and Credit River (43%), but a large area of Credit River is covered by loamy soil (49%). Grand River has almost an equal proportion of sand (30%), loam (32%) and clay (38%), while Thames River contains more clay (39%). The elevation is also highly variable with the highest altitudes in the North parts of Grand River (531 m) and Credit River (521 m) while the lowest areas are located in the sandplains further south in Grand River (178 m) and Big Creek (179 m).

## 2.2 PRMS hydrological model

The Precipitation Runoff Modelling System (PRMS), a semi-distributed conceptual hydrological model developed by Leavesley et al. (1983), was applied in all four watersheds to simulate the future evolution of streamflow for each member of a large climate ensemble. PRMS is used in this study because it needs only basic daily forcing climate data (minimum and maximum temperature, and precipitation). The advantage of using a model that need only few data as input is that it reduces uncertainties from multiple variables and reduce the model computational time. A drawback of using temperature is that energy balance is not physically represented. However, in an earlier study in the Big Creek watershed, PRMS represented well the snow processes (Champagne et al., 2019) showing that the use of temperature and precipitation are satisfactory to represent the snow processes in this region. Moreover, PRMS can be coupled with MODFLOW groundwater model (GSFLOW) to study the interaction between surface and groundwater flow (Markstrom and Regan, 2008). While MODFLOW was not activated in this study, having PRMS set up in these watersheds will facilitate the use of GSFLOW in future studies. This model has been widely applied in watersheds that experience periodic snowfall (Dressler et al., 2006; Liao and Zhuang, 2017; Mastin et al., 2011; Surfleet et al., 2012; Teng et al., 2017, 2018).

The hydrological calculations in PRMS are based on physical laws and empirical relations between measured and estimated quantities. A series of hydrologic reservoirs (plant canopy interception, snowpack, soil zone, subsurface) are used in the model and the water flowing between the reservoirs are computed for each grouped response units (GRUs). In this study the potential evapotranspiration was estimated using the Jensen-Haise formulation (Jensen and Haise, 1963). The interception was calculated separately for summer rain, winter rain and winter snow and was a function of the plant type. The separation between rainfall and snowfall was done by the snow module using temperature thresholds. If a day has a maximum temperature below $0°C$, all precipitation of the day was considered as snow. If a day has a minimum temperature higher than $0°C$ and a maximum temperature higher than a threshold to calibrate, then all precipitation is considered rain. A mixed precipitation is computed when conditions are between these values. The snowpack dynamics are simulated through estimate of energy and water dynamics. The energy available to melt the snow is based on estimation of shortwave radiation, longwave radiation, convection and condensation. Shortwave solar radiation was estimated using a degree-day method. Longwave radiation is the integration of the longwave radiation from the land cover and from the air depending on the emissivity of air. Convection and condensation are computed together as a function of temperature and a calibrated

coefficient. Surface runoff due to infiltration excess (Hortonian runoff) is computed using the antecedent soil moisture content. The amount of water not contributing to Hortonian runoff is infiltrated and directed to the soil zone. The soil zone module computes transpiration, recharge to the groundwater reservoir and three components of the streamflow: saturation excess (Dunnian runoff), subsurface flow through soil cracks, animal borrows or leaf litter (fast interflow) and subsurface

flow (slow interflow). These processes are described in more details by Markstrom et al., (2015)

The model was set up for each watershed using Arcpy-GSFLOW, a series of ARCGIS scripts (Gardner et al., 2018). Arcpy-GSFLOW constructed GRUs as surface grid cells of 200m² for Big Creek and Credit River watersheds and 400m² for Grand River and Thames River. These latter two larger watersheds have coarser GRU's because the parametrization with arcpy-GSFLOW is not functional with an excessive number of GRUs. An example of the GRUs grid is shown for the south part of

Big Creek (Supplementary material S1). Arcpy-GSFLOW calculated the physical characteristics of each GRU: Elevation, slope and aspect were derived from the High-Resolution Digital Elevation Model (HRDEM); The percentage of each land use type was derived from the Canadian Land Cover CIRCA 2000 (Natural Resources Canada, 2020) and used to calculate the rooting depth; and the Available water content, saturated hydraulic conductivity, and percentage of sand and clay were estimated using the materials from the surficial geology of Southern Ontario (Ontario Ministry of Northern development,

Mines and Forestry). From these calculated characteristics the spatialized parameters have been calculated at each GRU: The coefficients used to calculate slow interflow have been estimated using the saturated hydraulic conductivity and the slope. The maximum available water for plants was calculated using the available water content and the root depth and was used to estimate the total soil saturation. Finally, the linear coefficient used to route the water from the soil zone to the groundwater reservoir was estimated using the saturated hydraulic conductivity. The dominant land-use type (bare soil, grassland, shrubs,

coniferous trees or deciduous trees) and a single dominant soil type (sand, loam or clay) for each GRU were also estimated and used in some PRMS modules. Arcpy-GSFLOW was also used to define the stream network from the HRDEM. The accumulation flow threshold was determined empirically by matching the streams with aerial photographs. We then estimated the water cascade between the GRU's and the stream network. The lakes represent very small areas of the watersheds and therefore considered of negligible effect on streamflow. Control structures or dams were not taken into

consideration in this study because of their limited impact on the 30-years average streamflow used in this study. The model was calibrated and validated using the regulated flow series. Therefore, the dam effect should be implicitly accounted for during the model calibration and it is assumed that the reservoir levels will not change significantly in the future period.

The spatialized parameters estimated by Arcpy-GSFLOW were modified during calibration while keeping their relative spatial variability. Other parameters were lumped to the entire watershed and were calibrated as well (Table 2). Model

calibration was performed with a trial and error approach following three-steps: (1) The calibration of the daily shortwave radiation parameters using satellite data (2002-2008) from Natural resources Canada at 10km resolution (Djebbar et al., 2012); (2) The potential evapotranspiration (PET) parameters adjusted against PET values estimated using the Thornthwaite method (Thornthwaite, 1948) and (3) calibration of 17 parameters using the Normal Root Mean Square Error (NRMSE) between daily and monthly streamflow simulated by PRMS and daily and monthly observations measured at each watershed

outlet (blue triangles in Figure 1, Environment and Climate Change Canada Historical Hydrometric Data). A sensitivity analysis of the parameters in the Big Creek watershed (supplementary materials) shows that the infiltration to the soil zone is the most important process to accurately simulate the streamflow (smidx module). The available water threshold (soil_moist_max) as well as the travel time between stream segments (K_coef) are also important factors. For the snow module specifically, the convection/condensation energy coefficient (cecn) is the most sensitive (Section S3). The simulated streamflow was computed using precipitation, minimum temperature and maximum temperature from NRCANmet, the most commonly used dataset in Canada (Werner et al., 2019). The dataset was produced using station observation data from Environment and Climate Change Canada and Natural Resources Canada. The gridding at 10 km spatial resolution was accomplished using the Australian National University Spline (ANUSPLIN, McKenney et al., 2011). 186 data points were necessary to cover the area of the four watersheds. For model calculations, each GRU used climate data from the closest NRCANmet grid point (Section S1). Five years were used as the warm-up period (Oct 1984-Sept 1989) to remove any error due to initial conditions. Different simulations with a varying initialization period length were tested in the Big Creek watershed and showed that five years were necessary for the hydrological model to forget the initial conditions of the reservoirs. The calibration period was between Oct 1989 and Sept 2008 and the years 2009 to 2013 were used as the validation period.

The best sets of parameters retained after calibration are shown in Table 2. The spatial variability of the parameters estimated for each GRUs can be found in supplementary materials (Section S2). The Nash Sutcliff Efficiency (NSE) values are always higher than 0.65 for both calibration and validation periods (Table 3) which is generally considered a good quantitative fit (Moriasi et al., 2007). A percent bias (PBIAS) between -15% and +15%, also considered as a good fit was reached in our study with the exception of Credit River for the validation period. Figure 2 shows the simulation and the observation of the daily streamflow in all four watersheds and confirms visually the goodness of simulation fit. The ability of the best set of parameters to recreate the snow depth in the Big Creek watershed was tested in a previous study (Champagne et al., 2019) and shows good agreement with the observations.

**2.3 Climate data projections**

The set of parameters identified for each watershed during the calibration were used to simulate the future evolution of streamflow for each member of the Canadian Regional Climate Model large ensemble (CRCM5-LE). CRCM5-LE is a 50-member ensemble of climate change projections at 0.11° (~12-km) resolution available at 5-minute time steps over North-eastern North-America (Leduc et al., 2019). Each member of CRCM5-LE was driven by 6-hourly atmospheric and oceanic fields from each member of the Canadian Earth System Model version 2 large ensemble (CanESM2-LE) at a 2.8° (~310 km) resolution (Fyfe et al., 2017; Sigmond et al., 2018). The downscaling from CanESM2-LE was performed using the Canadian Regional Climate Model (CRCM5 v3.3.3.1; Martynov et al., 2010; Šeparović et al., 2013) developed by the ESCER Centre at UQAM (Université du Québec à Montréal) with the collaboration of Environment and Climate Change Canada. The ensemble extends from the historical (1954-2005) to the projected (2006-2099) period forced with the RCP8.5 scenario

(Meinshausen et al., 2011). The CRCM5-LE Data grid-points the closest to NRCANmet data points were used in this study. Before their use in PRMS, modelled temperature and precipitation from CRCM5-LE were bias-corrected monthly against NRCANmet at each grid point over the historical period (1954-2005). The intensity distribution of temperature was corrected using a normal distribution. For precipitation, a two-steps procedure was applied. The frequency distribution was first adjusted by truncating the modelled frequency. The truncated distribution of precipitation intensity was then corrected with a gamma distribution (Ines and Hansen, 2006). The bias correction method gives satisfactory results and was a necessary step before using CRCM5-LE in PRMS (Section S4). These bias-correction calculated from the historical period were applied at each CRCM5-LE grid point for the entire period 1954-2099.

## 2.4 Agglomerative hierarchical clustering

An agglomerative hierarchical clustering (AHC) was used to classify all 50 members into classes of similar change of forcing CRCM5-LE meteorological conditions and streamflow simulated by PRMS. AHC is a bottom-up clustering approach where each observation (here members) starts as its own cluster and one pair of clusters is merged at each step, respecting a minimum change of total variance between each step (Ward, 1963). In a general concept, the AHC calculates first the variance between each pair of observations. The pair with the lowest variance merges into a single class. In the next step, the pair of classes or pair of observations that would result in the smallest increase of total variance, compared to the previous step, is grouped together. This process is repeated until all classes of observations have been merged into a single class. In this study, the AHC was applied first to all four watershed's January-February normalized change in streamflow and then to the four watershed's average change of temperature and precipitation between the historical (1961-1990) and future (2026-2055) periods. The AHC was performed using January-February data because these months correspond to a large change in streamflow during the winter period. For precipitation and temperature, the period from 25 December to 22 February was used to account for the delay between weather conditions and streamflow at the outlet. A delay of six days showed the best correlation between the increase in temperature and precipitation and the increase in streamflow for all four watersheds. The number of classes to retain for change of streamflow and number of classes for change of weather conditions correspond to the highest change in variance. The classification was used here to simplify the study of the connections between the future change in large scale atmospheric circulation, local meteorological conditions and streamflow. This method using streamflow response classification rather than using a classification of climatological patterns was chosen because is focusing on the impact that can be used in other hydrological application.

The future projection of atmospheric circulation for each class was analysed using climate variables from CanESM2-LE with a geographical domain from 30°N to 60°N latitude and 100°W to 50°W longitude. Climate variables used for analysis included air temperature at 850hPa level (850T), precipitation (PP), sea level pressure (SLP), geopotential height at 500hPa (Z500) and surface winds. These climate variables were separated into internal and forcing contributors. The forcing contribution of the climate variables corresponds to the average change of all ensemble members between the historical period and future simulations. The internal contribution associated with each member was calculated by subtracting the

original member data from the forcing contribution. This method was previously used by Deser et al. (2014) to assess the internal contribution of future change in temperature and precipitation in North America.

## 3 Results

### 3.1 Streamflow projections

Figure 3 shows the average daily streamflow volume and the number of high flows for all members for the historical (HIST) and future (2040s) periods. Observational streamflow measured at each watershed outlet (OBS) and the streamflow simulated by PRMS using observed temperature and precipitation from NRCANmet (CTL for control) are also shown for the historical period. A day is considered a high flow when the streamflow value is higher than the mean plus 3 times the standard deviation, based on observed streamflow. When at least two days in a row satisfy this condition, only one day of the

series is considered as a high flow.

In the historical period, average streamflow from OBS, CTL and the 50-member data sets followed similar annual cycles with the first peak of the hydrological year occurring in November-December and the highest peak in March-April. By 2040, a clear peak in streamflow and number of high-flow events are still modelled in March but streamflow is more evenly distributed among winter months. This result suggests a shift from two maximal peaks to one winter peak by the mid-21st

Century. The largest increase in streamflow occurred in January-February with a 50-members average increase reaching 18% (±8.7) in Big Creek, 30.5% (±10.8) in Grand River, 29.8% (±10.4) in Thames River and 31.2% (±13.3) in Credit River. All 50 members depict a streamflow increase in winter, but the simulated range of streamflow volume and the number of high flows is wide among the 50 different members.

Daily rainfall, snowmelt, and actual ET are also expected to change by 2040s (Figure 4). The amount of rain is simulated to

consistently increase among the 50-member average in winter and early spring in all four watersheds. The 50-members average November-April increase in rainfall is about 29.7% (±8.7) in Big Creek, 37.3% (±10.3) in Grand River, 30.7% (±8.6) in Thames and 40.3% (±11.7) in Credit River. In summer, PRMS simulates future average rainfall to decline between 5 and 8.5% depending on the watershed, but the direction of change is inconsistent between individual members. The amount of snowmelt is expected to shift from high melt volume in March to a volume consistent throughout the winter. In

March-April, snowmelt is expected to decline by 61.9% (±11.2) in Big creek, 52.2% (±10.7) in Grand River, 60.5% (±10.5) in Thames River and 42.8% (±11.8) in Credit River, while in January-February, snowmelt is expected to increase by 10.2% (±12.5) in Big creek, 32.2% (±12.7) in Grand River, 23.7% (±11.7) in Thames River and 45.8% (±16.1) in Credit River. Future ET will slightly increase for most months but decrease in summer.

Figure 5 shows the 50-member historical and projected bias-corrected temperature and precipitation for all four watersheds.

Air temperature is shown to consistently increase for all months while the range of precipitation amounts projected by the 50 members is wider compared to the change in temperature. On average, simulated precipitation increases in November-April and decreases in June-September.

### 3.2 January-February streamflow projections variability

The 50 members of the ensemble were classified first in classes of similar January-February streamflow change between the historical period and 2040s using the AHC described in the method section. The number of classes to retain was determined using a dendrogram (Figure 6). The dendrogram shows the cumulative total intraclass variance of normalized streamflow change for the successive merging, from the first merging that uses all members (bottom) to the last merging creating a single class (top). The highest vertical distance between two successive merging in the Y axis corresponds to the change in number of classes affected by the highest intraclass variance increase. The number of weather classes was identified using the same method (Figure 6). Three streamflow classes (HiQ, MoQ and LoQ for high, medium and low increase of streamflow) and four weather classes (HiPT, MoPT, LoPT and HiT) correspond to the classes merged right before the highest change in variance (Figure 6). Three of the weather classes (HiPT, MoPT and LoPT) show a gradient from high to low increase for both precipitation and temperature while one weather class show a high increase in temperature but low increase in precipitation (HiT) (Figure 6, right panel). The labels High and Low are not refering to absolute values but correspond to higher or lower increase in streamflow and temperature/precipitation relative to the other members.

The streamflow and weather classes were then aggregated, grouping the members that are in the same streamflow and weather classes, giving a total of nine classes (Table 4). The increase in streamflow is similar between watersheds with the exception of Big Creek depicting a lower change. In Big Creek the classes corresponding to HiQ have an average increase comprises between 25% and 32%, MoQ between 18% and 24% and LoQ between 8% and 14%. In the three other watersheds HiQ depicts an average increase comprises between 39% and 54%, MoQ between 28% and 36% and LoQ between 18% and 24% (Table 4). The interclass variability is also generally consistent between watersheds with the exception of Big Creek where the classes HiQHiT and LoQHiT show a comparatively low streamflow increases as compared to other classes (Table 4).

The table 4 emphasized that despite a simlar change in precipitation and temperature, the streamflow varies greatly between classes. Figure 7 shows scatter plots of averaged change of streamflow to average change of precipitation, temperature, snowmelt and rain between the historical period and the 2040s period for all nine classes shown in Table 4. HiQHiPT and LoQLoPT classes are associated with the highest (lowest) increases of streamflow due to high (low) increases of snowmelt and rain (Figure 7). The larger increase in rain and snowmelt for HiQHiPT members are likely due a larger warming and increase in precipitation. MoQLoPT demonstrates a larger increase in simulated streamflow compared to LoQLoPT, which is likely due to a larger increase of precipitation amounts despite lower warming. MoQLoPT is especially larger than LoQLoPT in term of snowmelt suggesting more snowfall for MoQLoPT members. The three weather classes associated with a large increase of temperature only (HiT) depict a moderate increase of rain and snowmelt suggesting that these members increase the rain to snow ratio and accelerate the snowmelt. LoQHiT also shows a strong warming but a low increase of snowmelt explaining the low increase in streamflow (Figure 7). Lastly, MoQMoPT has a higher increase in both rainfall and snowmelt compared to LoQMoPT, but both classes demonstrate similar change of precipitation and temperature. These

results suggest that alternative factors than average change in temperature and precipitation could explain the change in rainfall, snowmelt and streamflow in january-february. These factors will be described in part 3.4 and discuss in section 4.4. Lastly, the main visual difference between watersheds was a lower increase of snowmelt expected in Big Creek.

### 3.3 Atmospheric circulation and streamflow projections

The 50 members average change of temperature and precipitation between the historical period and the 2040's is shown in Figure 8. An increase of air temperature at 850hPa (T850) and geopotential height at 500hPa (Z500) is expected to occur within the entire domain with a stronger gradient closer to the Arctic (Figure 8c). Precipitation is also simulated to increase by the 2040s throughout the domain while SLP is expected to decrease (Figure 8d). In the region close to the Great Lakes, the magnitude of warming and variability between members is higher on the northern shorelines as compared to the open water and shorelines south of the Lakes (Figure 8a). Precipitation increases is also projected to be higher on land and on the east side of the Great Lakes and toward the Atlantic coast (Figure 8b and 8d).

The internal contribution of each CanESM2-LE member to the change of climate variables was averaged for each class (Figure 9). The class HiQHiPT is projected to be associated with positive temperature, precipitation, and southwesterly winds change anomalies between high pressure anomalies in the east and low pressure anomalies in west side of the domain (Figure 9a and 9h). LoQLoPT has opposite pressure gradient anomalies and is the only class that show negative precipitation and temperature change anomalies occurring simultaneously (Figure 9g and 9n). LoQMoPT demonstrates a similar pattern to LoQLoPT, but the negative pressure anomalies are attenuated, and precipitation increase is higher (Figure 9e and 9l). MoQHiT and LoQHiT are characterized by positive temperature and pressure change anomalies over southern Ontario, while MoQMoPT and MoQLoPT have an opposite pattern.

### 3.4 Antecedant conditions and streamflow

Alternative factors than January-February atmospheric conditions are also examined that may help to explain the January-February evolution of streamflow between the historical and the future period. Figure 10 shows the change of precipitation amount in November-December, groundwater flow in January-February and amount of snowpack water equivalent for the first and the last day of the January-February period.

November-December precipitation are projected to increase for all classes but a large intraclass and interclass variability is shown. The classes HiHiPT, HiHiT, MoHiT and the two LoPT weather classes show visually a higher increase of November-December precipitation as compared to the other classes. The amount of snowpack water equivalent at the beginning of the January-February period is expected to decrease with low variability between the classes but a large intra-class variability (Figure 10). The snowpack at the end of January-February is expected to decrease significantly for all classes with a low intraclass variability. The groundwater flow shows visually a lower increase in Big Creek compared to the other watersheds likely due to a lower overall increase in streamflow.

## 4 Discussion

### 4.1 Historical simulations

The observed seasonal cycle of streamflow was visually well reproduced by the simulated CTL and ensemble data for the historical period (1961-1990) (Figure 3). However, the simulated streamflow from CTL and the ensemble overestimated streamflow between November and February in the Thames and Big Creek watersheds. The overestimation is stronger in January for the ensemble which can be attributed to an overestimation of precipitation (Figure 5). Winter overestimation was previously reported for the Grand River watershed (Erler et al., 2018) and was attributed to the lack of ponding or frozen soil process representation in the model. Similarly, the version of PRMS used in our study did not represent the ponding and frozen soil processes. However, a comparison of the observed streamflow during frozen and non-frozen soil periods in the Big creek watershed showed a small difference (figure not shown) suggesting a small impact of frozen soil on the streamflow in this region. Moreover, the streamflow simulations using NRCANmet data performed well in Grand River (Figure 3). These results suggest that other factors than the hydrological model are likely responsible for the discrepancies in Thames River and Big Creek. The quality of NRCANmet observations could also be a source of uncertainty. The ANNUSPLIN method, used by NRCANmet to interpolate the station-based observations, generally overestimates precipitation in this region (Newlands et al., 2011). Despites these biases, NRCANmet is among the most widely used gridded dataset in Canada (Werner et al., 2019) and the use of NRCANmet to simulate snow processes was satisfactory in the Big Creek watershed (Champagne et al., 2019). The observed streamflow itself can also be affected by measurement uncertainty during ice conditions and especially an overestimation of the discharge. The validation of simulations using other variables such as evapotranspiration or soil moisture would be beneficial to improve the confidence in the results. Evapotranspiration from CRCM5-LE was not available for this work but could be investigated in future works.

### 4.2 Increase in streamflow amplified or attenuated by Z500 anomalies

Despite the discrepancies highlighted in the last section, the results show a clear increase of streamflow in January-February (Figure 3) which has been previously simulated for other watersheds in the Great Lakes region (Byun et al., 2019; Erler et al., 2018; Grillakis et al., 2011; Kuo et al., 2017). January-February streamflow increases will likely be caused by temperature and precipitation increases (Figure 5 and 8) that causes rain and snowmelt amounts to rise (Figure 4). Grillakis et al., (2011) used several hydrological models in a small catchment close to Lake Ontario and projected streamflow increases due to rainfall increases in January and snowmelt increases in February. In our study we found an increase of rain and snowmelt for both months (Figure 4). The future increase of January-February rain and snowmelt can be associated with the warming simulated by CanESM2-LE (Figure 8). This warming has a similar amplitude compared to other CMIP5 model projections forced with the same RCP8.5 scenario (Zhang et al., 2019). An increase in January-February precipitation, projected in a large part of the domain (Figure 8), is also similar to other climate models simulations (Zhang et al., 2019). Precipitation increase between Lake Ontario/Erie and the East coast (Figure 8) is not expected by the CMIP5 multi-model

projections and is likely inherent to CanESM2-LE. This precipitation pattern is probably associated with stronger winds from the east coast (Atlantic Ocean) due to a higher pressure decrease on land (Figure 8).

The 50 members produce a variable increase of streamflow (Figure 3) which is likely due to the variability in atmospheric circulation (Figure 9). 14% of the ensemble showed a high increase of streamflow simultaneously with high geopotential height anomalies near the east coast and southerly winds through the Great Lakes region (Table 4 and Figure 9a and 9h). High geopotential height anomalies located in the eastern United States has been previously found responsible for more precipitation and higher temperature in the Great Lakes region in winter (Mallakpour and Villarini, 2016; Thiombiano et al., 2017), thereby increasing the streamflow and high flow events (Bradbury et al., 2002; Mallakpour and Villarini, 2016). 14% of the ensemble corresponds to the opposite pattern with low geopotential height anomalies in the east coast and northern winds anomalies (Figure 9g and 9n). These atmospheric conditions will attenuate the warming and precipitation amounts and will be therefore associated with a lower increase of streamflow (Table 4 and Figure 7). 6% of the ensemble (Class MoQLoPT) shows a low warming but a moderate increase in precipitation and snowmelt (Figure 7 and 9f and 9m) suggesting snowfall enhancement. The north-west wind anomalies associated with this class (Figure 9f and 9m) could enhance snowfall in this region through lake effect snow (Suriano and Leathers, 2017). Another 16% of the ensemble shows a moderate increase in streamflow associated with a strong warming (MoQHiT) which may be driven by high-geopotential height anomalies on the Great Lakes (Figure 9b and 9i). This pattern drove moderate increases of snowmelt and rain-to-snow ratio associated with strong warming (Figure 7, 9b and 9i). Correspondence between high geopotential height and high temperature on the Great Lakes in winter have been previously reported (Ning and Bradley, 2015). Ning and Bradley (2015) suggested that the high geopotential anomalies on the Great Lakes prevent the polar jet-stream and the cold air masses from entering the region.

## 4.3 Consistency in the weather classes

The weather classes are associated with specific trends in atmospheric conditions (Figure 9) but are composed from an average of members that have their own atmospheric signature despite a similar impact on local conditions. Changes in Z500 and T850 anomalies for each member are depicted in Figure 11 to investigate the atmospheric variability between members. The members that comprise classes HiPT show a large increase in Z500 anomalies in the east coast consistently for six members while for two members (#13 and #48) the high increase in Z500 anomalies is centered north from the Great Lakes. Eight members of the class LoPT show strong Z500 decrease in the east coast but in two members (#1 and #10) the decline is rather centered in the northern side of the Great Lakes. HiT show generally Z500 increase centered on the Great Lakes but four of the thirteen members depict a different pattern (#2, #20, #31 and #47). Finally, members from MoPT show generally a decrease in Z500 but we observe a high diversity in the change of circulation patterns. Members from MoPT depict a lower Z500 gradient compared to other classes suggesting a lower contribution of internal variability of climate to the total change in atmospheric conditions (Figure 11). These results suggest a large variability in atmospheric circulation change between members of the same ensemble with some members showing very unique change in atmospheric circulation. Despite the

atmospheric anomalies differences between members predicting similar local weather, the classes method used in this study gives a good probabilistic overview on how the change in regional atmospheric anomalies will impact local weather.

## 4.4 Lag between atmospheric circulation shifts, local climate conditions and streamflow

Results show that interclass variability in the increase of January-February streamflow is mostly due to temperature and
precipitation variability in the same months. The members with the highest increase in January-February precipitation and temperature (HiPT) are the members associated with the highest January-February streamflow increases, except for MoQHiPT (Table 4). The members associated with the lowest increase in precipitation and temperature (LoPT) show the lowest streamflow increase (LoQLoPT). Three other members of LoPT are associated with higher streamflow increase (MoQLoPT) which can be due to more precipitation and snowfall despite a lower warming (Figure 7).
Within the other two weather classes, HiT and MoPT, a similar change in January-February weather conditions translates to a large range of streamflow projections. These discrepancies between the evolution of weather conditions and streamflow volume in January-February can be associated with a delay between weather conditions and streamflow. To account for the routing delay between rain/snowmelt events and streamflow observed at the outlet, our analyses used a lag-time of six days between the precipitation/temperature and the streamflow. Any remaining delay between weather conditions and streamflow
could occur due to snowpack remaining from the previous months. Figure 10 shows a low variability between all MoPT members and all HiT members in term of change in starting snowpack volume suggesting a low impact of snowpack remaining at the end of December on change in January-February streamflow. Meanwhile, snowpack remaining at the end of January-February is decreasing at a higher rate for MoQMoPT members as compared to LoQMoPT members and for MoQHiT members compared to LoQHiT members (Figure 10) which may be associated with a higher increase in snowmelt
(Figure 7). However, these two classes show very similar change of temperature and precipitation (Figure 7) suggesting that average weather change obscures intra-seasonal variability change. For example, if more snow falls in the second half of February and temperature stays below the freezing point, this snow is likely to melt in March and is therefore not counted in the January-February streamflow.

The discrepancy between change in weather conditions and streamflow can also be due to groundwater recharge/discharge
variability. The lower streamflow increase in LoQHiT is for example associated simultaneously with a lower increase in groundwater flow and a lower increase in November-December precipitation amount (Figure 10). A correlation close to 0.7 between the 50 members November-December change in precipitation amount and the January-February change in groundwater flow confirms the connection between fall precipitation and winter groundwater flow. The processes connecting fall precipitation and winter groundwater will need further investigation with the help of a coupled surface and
groundwater model, such as GSFLOW for instance (Markstrom et al., 2015). These results emphasize the possible role of processes delaying the streamflow (i.e. Snowpack, Groundwater) and the need to also study the succession of different atmospheric patterns in the previous months before the January-February modulation of streamflow.

## 4.5 Spatial variability of streamflow change modulation

The changes in the amount of rain and snowmelt between the historical period and the 2040's are visually similar for three of the watersheds (Figure 7). The Big Creek watershed is distinctly different as it shows a lower snowmelt contribution to streamflow (Figure 7). This suggests a thinner snowpack available for melting in this watershed as it is situated in the southern part of the study area near Lake Erie and experiences the mildest winters (Figure 5). In this watershed, the snowmelt volume is expected to increase only slightly in January (Figure 4). The increase in snowmelt is also expected to occur only in January for Thames River while the increase will be stronger in February for Grand and Credit River. A similar South-North pattern is observed in previous studies. A high increase in streamflow in December and January followed by a decrease in streamflow in February was simulated for the Canard watershed near Lake Erie (Rahman et al., 2012) while this shift is expected to occur between February and March further north near Lake Ontario (Grillakis et al., 2011; Sultana and Coulibaly, 2011) or Lake Simcoe (Kuo et al., 2017; Oni et al., 2014). These results suggest that the winter increase in streamflow is expected to be lower in the warmest watersheds classically situated further south, in lowlands and close to the Great Lakes. In these watersheds the snowpack was already reduced in the historical period and the further warming is not expected to increase the snowmelt contribution to the streamflow. However, similar to previous studies in southern Ontario, the reduced snowpack is not projected to decrease the streamflow in winter because the winter precipitation are also projected to increase as suggested in the majority of the climate models (Zhang et al., 2019).

## 5 Conclusion

This study used a 50-member ensemble of regional climate data, forced with the IPCC RCP8.5 scenario, as input in the PRMS hydrological model to show how the internal variability of climate is transferred to the near future winter (January-February) projections of streamflow in four diverse watersheds in southern Ontario, Great Lakes region. An agglomerative hierarchical clustering method was used to construct classes of similar change in temperatures/precipitation/streamflow and define streamflow change probabilities and associated regional atmospheric drivers. First, the results showed that all members of the ensemble were associated with a January-February increase in streamflow between 1961-1990 and 2026-2055, with an average increase of 18% (±8.7) in Big Creek, 30.5% (±10.8) in Grand River, 29.8% (±10.4) in Thames river and 31.2% (±13.3) in Credit River. This streamflow increase is due to a strong warming trend and an increase in precipitation projected by the IPCC RCP 8.5 scenario. Second, the results suggested that the future increase of temperature and precipitation in January-February will be modulated by the internal variability of climate with implication for hydrological processes. Specifically, our study showed that:

(i) One class of CRCM5-LE members, representing 14% of all ensemble members, depicted an amplification in the future average streamflow. The average streamflow change for this class will be as high as +31.6% (±8.1) in Big Creek, +48.3% (±11.1) in Grand River, +47% (±9.6) in Thames river and +53.7% (±15) in Credit River.

This amplification will be due to rainfall and snowmelt enhancement associated with the development of high-pressure anomalies in the east coast of North America.

(ii)    The opposite pattern, associated with anomalous low pressure in the east coast of north America, showed an attenuation in average streamflow. This class depicted a future change in streamflow of only +8.3% ($\pm$7.8) in Big Creek, +18.8% ($\pm$5.8) in Grand River, +17.8% ($\pm$6.4) in Thames river and +18.6% ($\pm$6.5) in Credit River.

(iii)    Two other classes representing another 24% of all ensemble members showed a moderate attenuation in streamflow increase with +12.7% ($\pm$3.6) in Big Creek, +22.3% ($\pm$3.3) in Grand River, +23% ($\pm$2.3) in Thames river and +21.1% ($\pm$6) in Credit River. This attenuation might occur due to low November-December precipitation and low January-February snow accumulation/melting.

(iv)    Almost half of all ensemble members showed a change in temperature and precipitation close to the 50-members average and showed a small contribution of internal variability of climate to the projected variability of streamflow.

These results focussing on average change of atmospheric conditions cannot be applied to high flows, mostly driven by the day to day variability of atmospheric circulation. The use of the same regional ensemble together with a classification of daily atmospheric fields would be useful to assess the future projections of high flows and flood regimes in the region. Despite a large number of regional climate simulations used here to drive a hydrological model, the results are derived from a single model chain (CanESM2, CRCM5 and PRMS). As a result, this ensemble does not consider other important sources of uncertainty from emission scenario and model structure. Future studies could use other global climate models and different scenarios and can be extended to the end of the 21st century. Other hydrological models could also be used to increase the confidence regarding the projections of hydrological processes. This work is important to assess the natural variability of the hydrological projections and help the society to be prepared for large range of future changes in flooding regimes.

**Data availability**

The historical hydrometric data can be extracted from the Environment and Climate Change Canada Historical Hydrometric Data website (https://wateroffice.ec.gc.ca/mainmenu/historical_data_index_e.html, last access: 3 February 2020) (Environment and Climate Change Canada, 2020). PRMS model codes are accessible from the USGS website (https://www.usgs.gov/software/precipitation-runoff-modeling-system-prms, last access: 24 March 2020) (USGS, 2020). CRCM5-LE data are not publicly available. Martin Leduc should be contacted for any request (leduc.martin@ouranos.ca). Model simulations and sequences of weather regimes are available upon request from M. Altaf Arain (arainm@mcmaster.ca).

**Authors contribution**

ML furnished CRCM5-LE data. OC prepared the figures and performed the analyses. OC prepared the manuscript with contributions from all co-authors.

**Competing interests**

The authors declare that they have no conflict of interest.

**Acknowledgments**

Financial support for this study was provided by the Natural Sciences and Engineering Research Council (NSERC) of Canada through the FloodNet Project. The production of ClimEx was funded within the ClimEx project by the Bavarian State Ministry for the Environment and Consumer Protection. The CRCM5 was developed by the ESCER centre of

Université du Québec à Montréal (UQAM; www.escer.uqam.ca) in collaboration with Environment and Climate Change Canada. We acknowledge Environment and Climate Change Canada's Canadian Centre for Climate Modelling and Analysis for executing and making available the CanESM2 large ensemble simulations used in this study, and the Canadian Sea Ice and Snow Evolution Network for proposing the simulations. Computations with the CRCM5 for the ClimEx project were made on the SuperMUC supercomputer at Leibniz Supercomputing Centre (LRZ) of the Bavarian Academy of Sciences and

Humanities. The operation of this supercomputer is funded via the Gauss Centre for Supercomputing (GCS) by the German Federal Ministry of Education and Research and the Bavarian State Ministry of Education, Science and the Arts. We also acknowledge Natural Resources Canada for their contribution in providing climate data sets, Global Water Future Program for their support and the reviewers for their constructive comments on the paper.

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

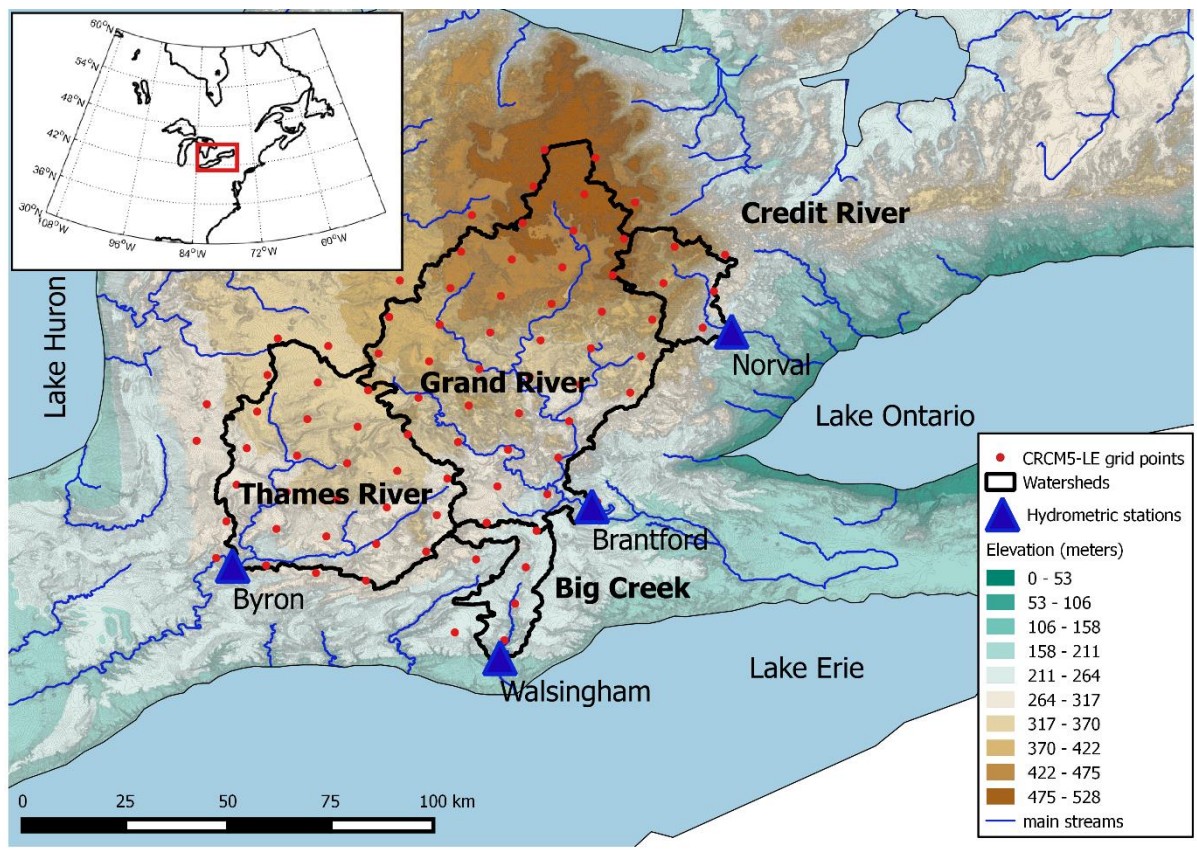

**Figure 1: Location map of the four studied watersheds in Southern Ontario. Elevation source: High Resolution Digital Elevation Model (HRDEM, Natural resources Canada).**

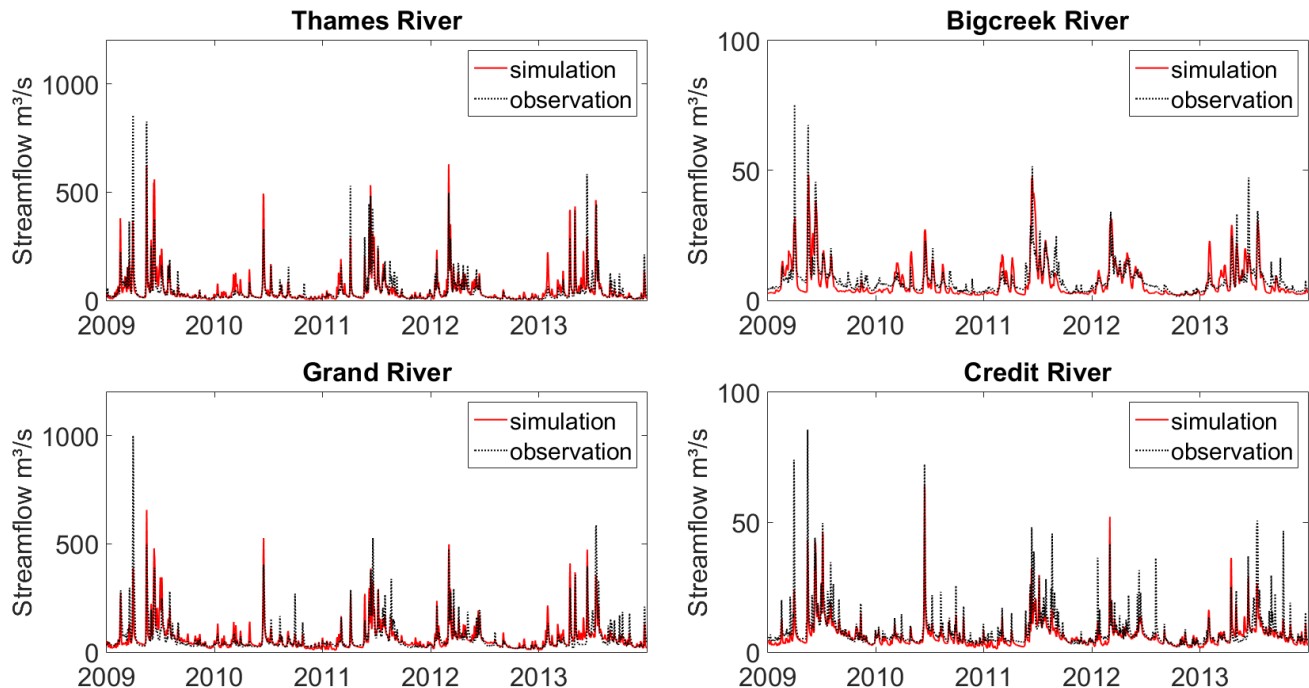

**Figure 2: Daily observed (OBS) and simulated (CTL) streamflow during the validation period (2009-2013).**

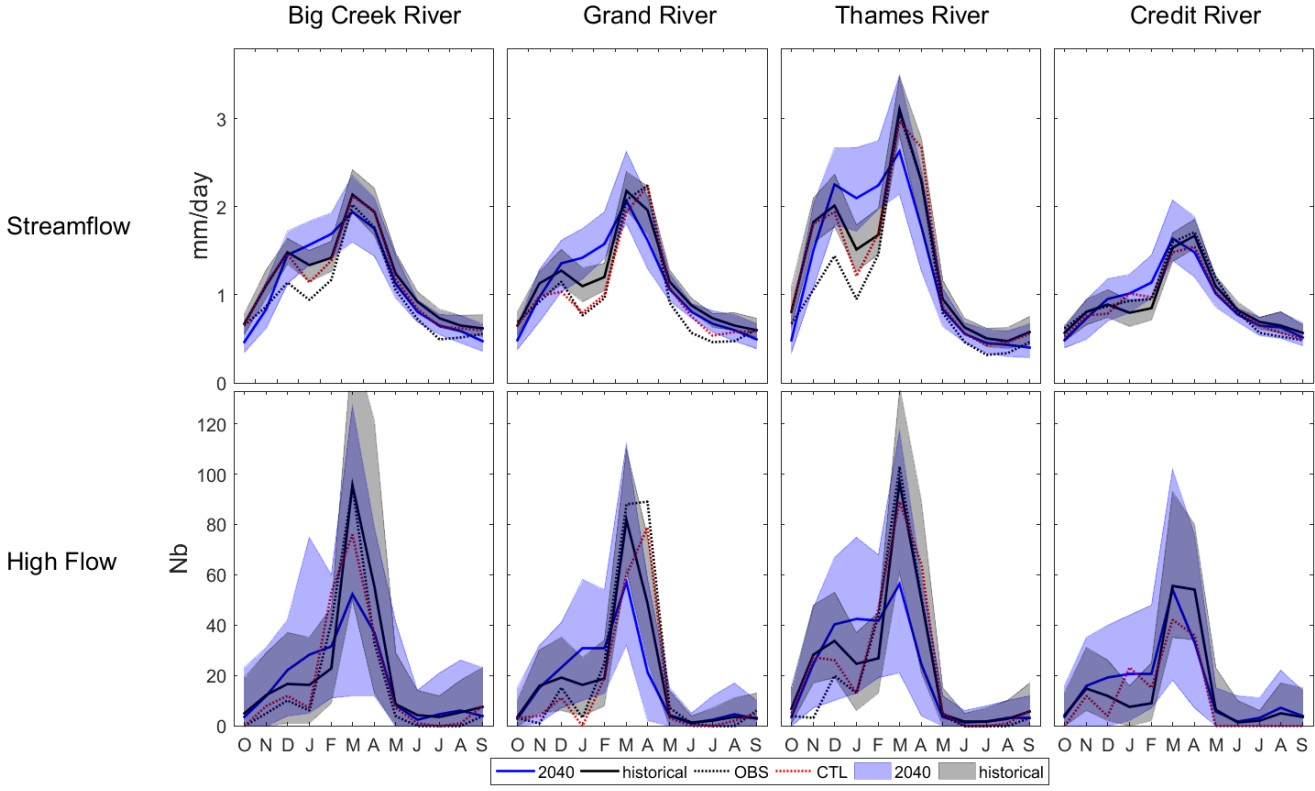

**Figure 3: 50-members range and average streamflow and number of high-flows for the historical and the 2040's periods.**

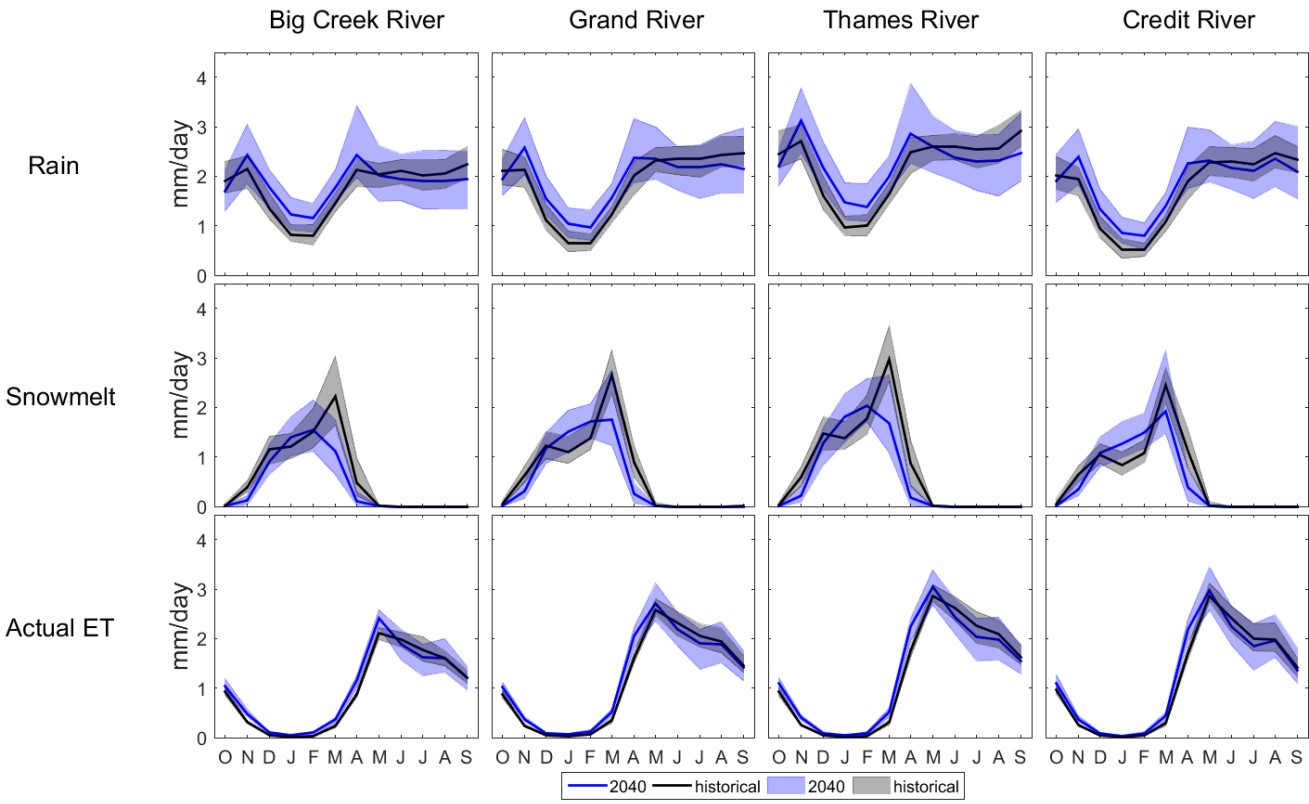

**Figure 4: 50-members range and average rain, snowmelt and actual ET amounts for the historical and the 2040's periods.**

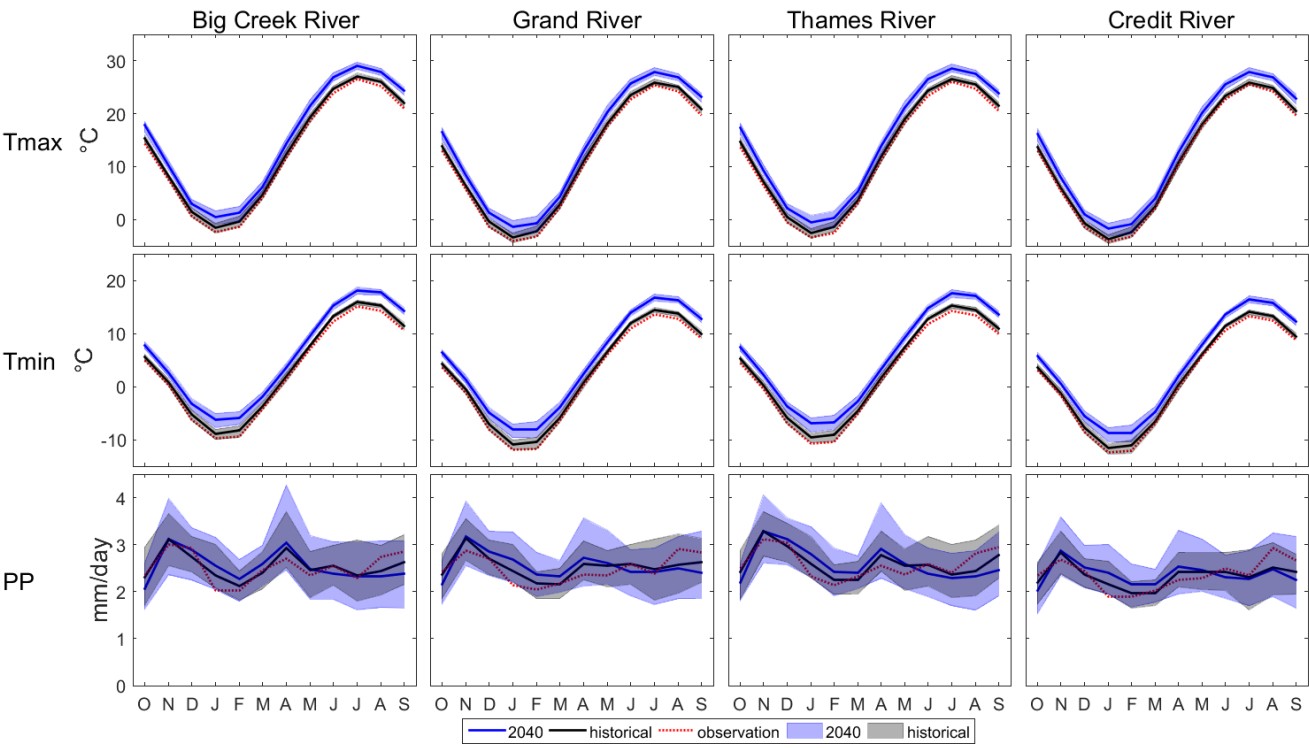

**Figure 5: CRCM5 50-members range and average bias-corrected temperature and precipitation amounts for the historical and the 2040's periods, together with the observed temperature and precipitation.**

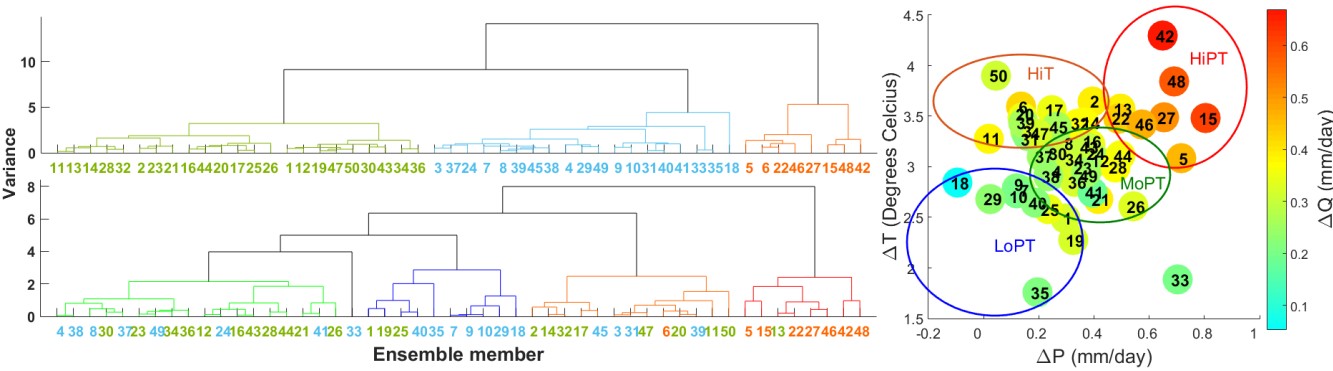

**Figure 6 Left:** Results of the Agglomerative hierarchical clustering (AHC) for the normalized change of streamflow (Q) (above) and normalized change of average Temperature (T) and Precipitation (P) (below). Colored numbers represent Q classes. **Right:** 4-watersheds average change of streamflow (Q) (Colors) with respect to average change of P and T. Large hollow circles represent the 4 weather classes.

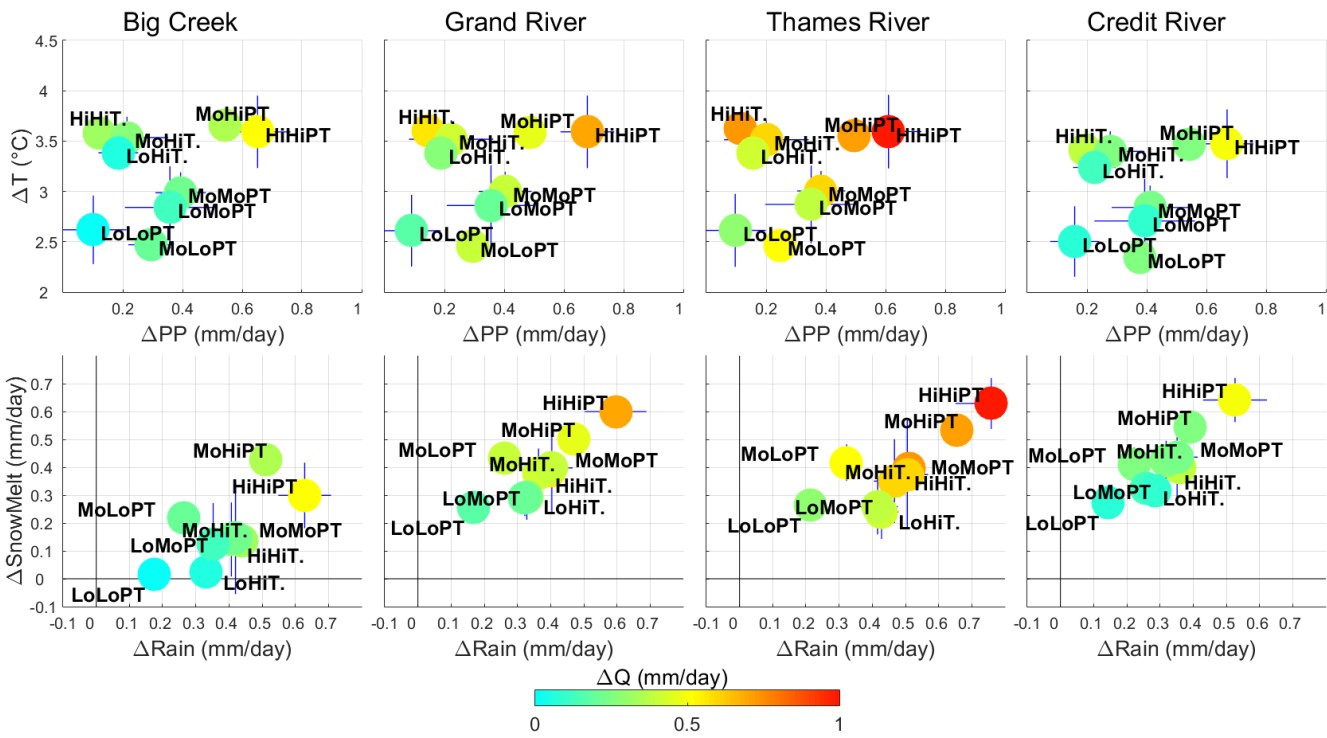

**Figure 7: Change of streamflow (Colors) with respect to changes of daily temperature and precipitation amount (above) and snowmelt and rain amounts (Below) between the historical and the 2040's future period in January-February.**

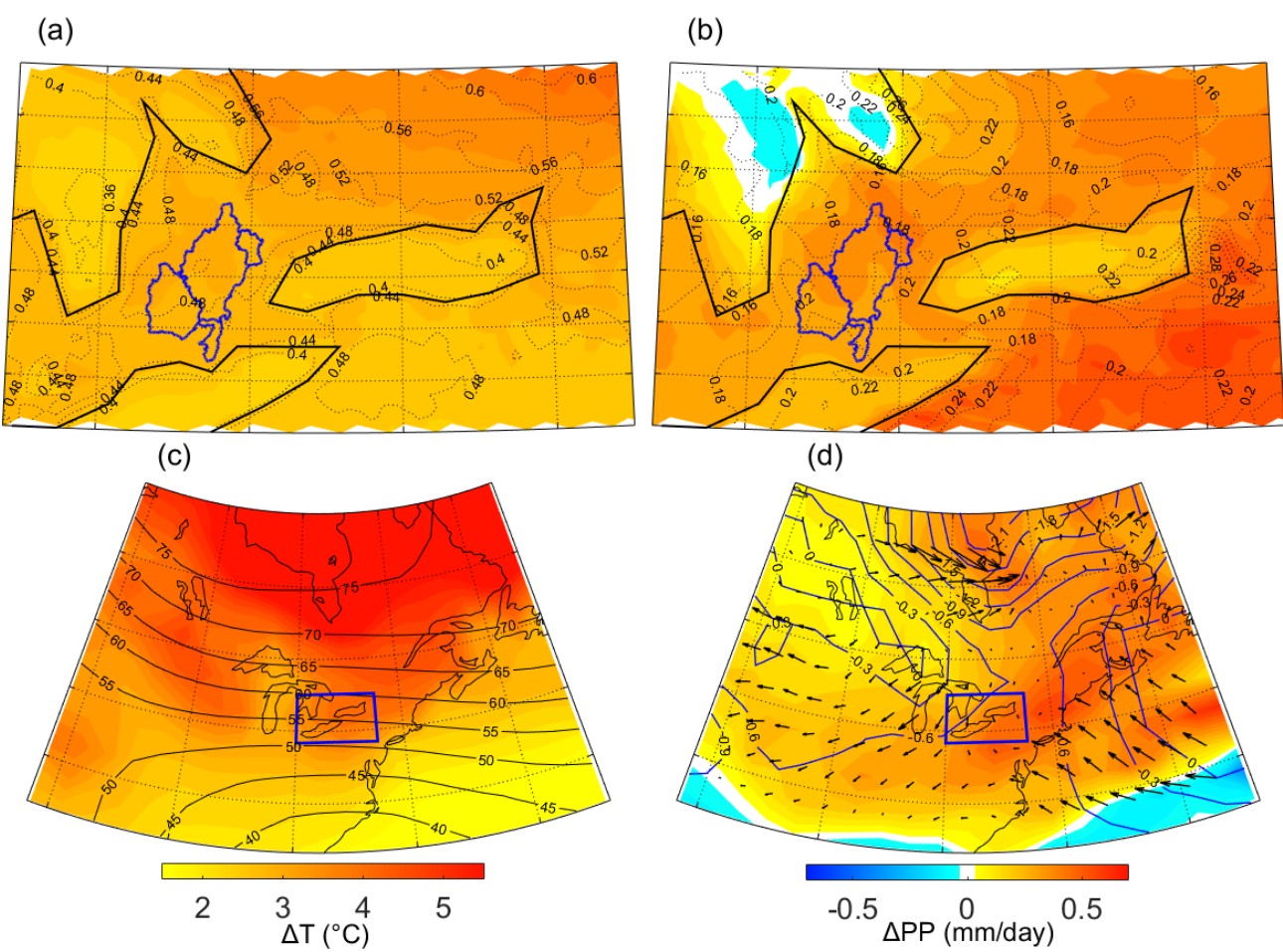

**Figure 8.** 50-members ensemble average change of atmospheric conditions between the historical and the 2040's period in January-February for a. CRCM5-LE average surface temperature (shade) and standard deviation (black lines), b. CRCM5-LE average daily precipitation (shade) and standard deviation (black lines), c. CanESM2-LE T850 (shade) and Z500 (black lines) and d. CanESM2-LE daily precipitation (shade), SLP (blue lines) and wind (vectors).

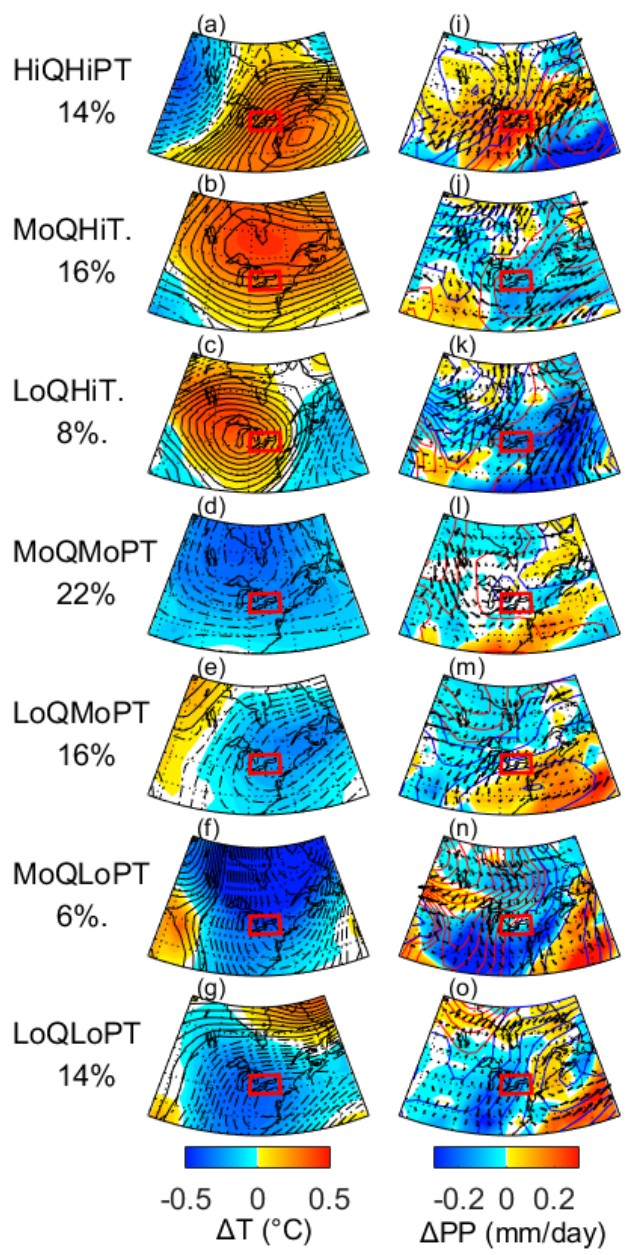

**Figure 9: a-g: Classes averaged internal contribution of a-g T850 (shade) and Z500 (black lines, in intervals of 1m) and h-n: Precipitation (shade), SLP (lines, in intervals of 0.1Pa) and wind (vectors) to the 50-members average change between the historical and the 2040's period in January-February.**

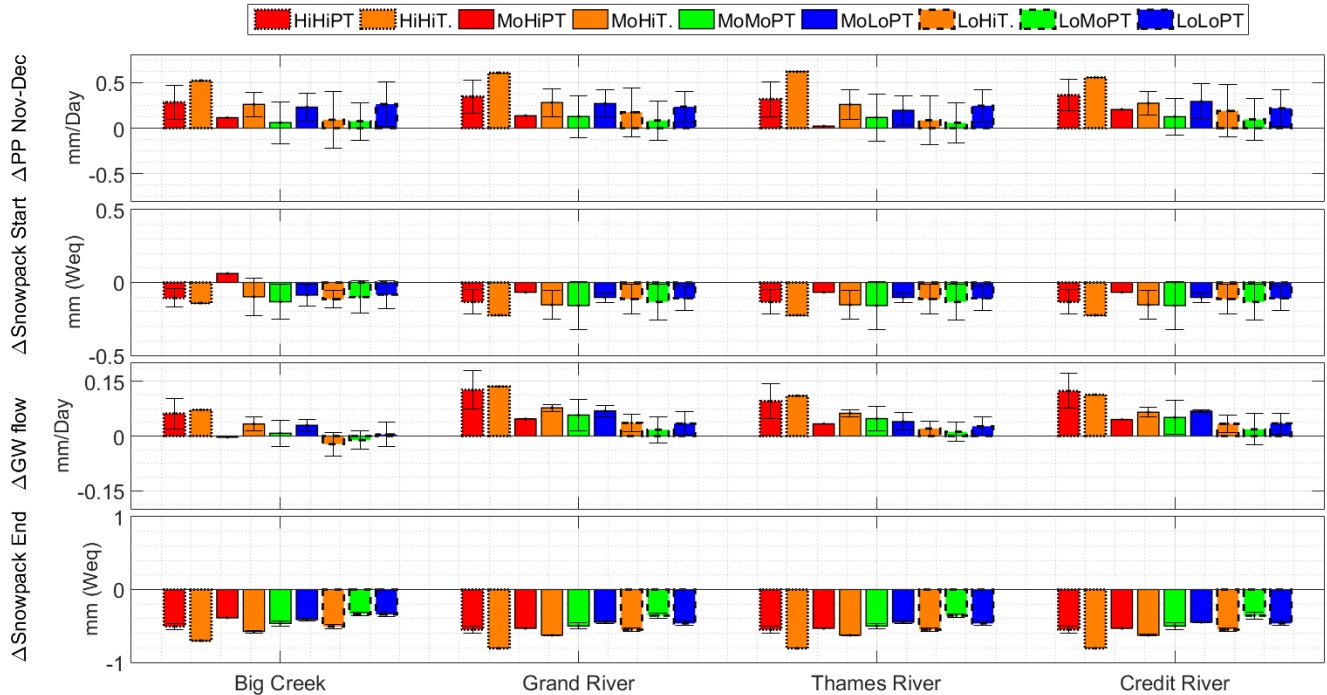

**Figure 10: Classes average (bars) and standard deviation (hatches) of the change between the historical and 2040's period of precipitation amount (mm) in November-December, snowpack amount (mm water-equivalent) in December 25th, Groundwater flow in January-February, and snowpack amount (mm water-equivalent) in February 23th.**

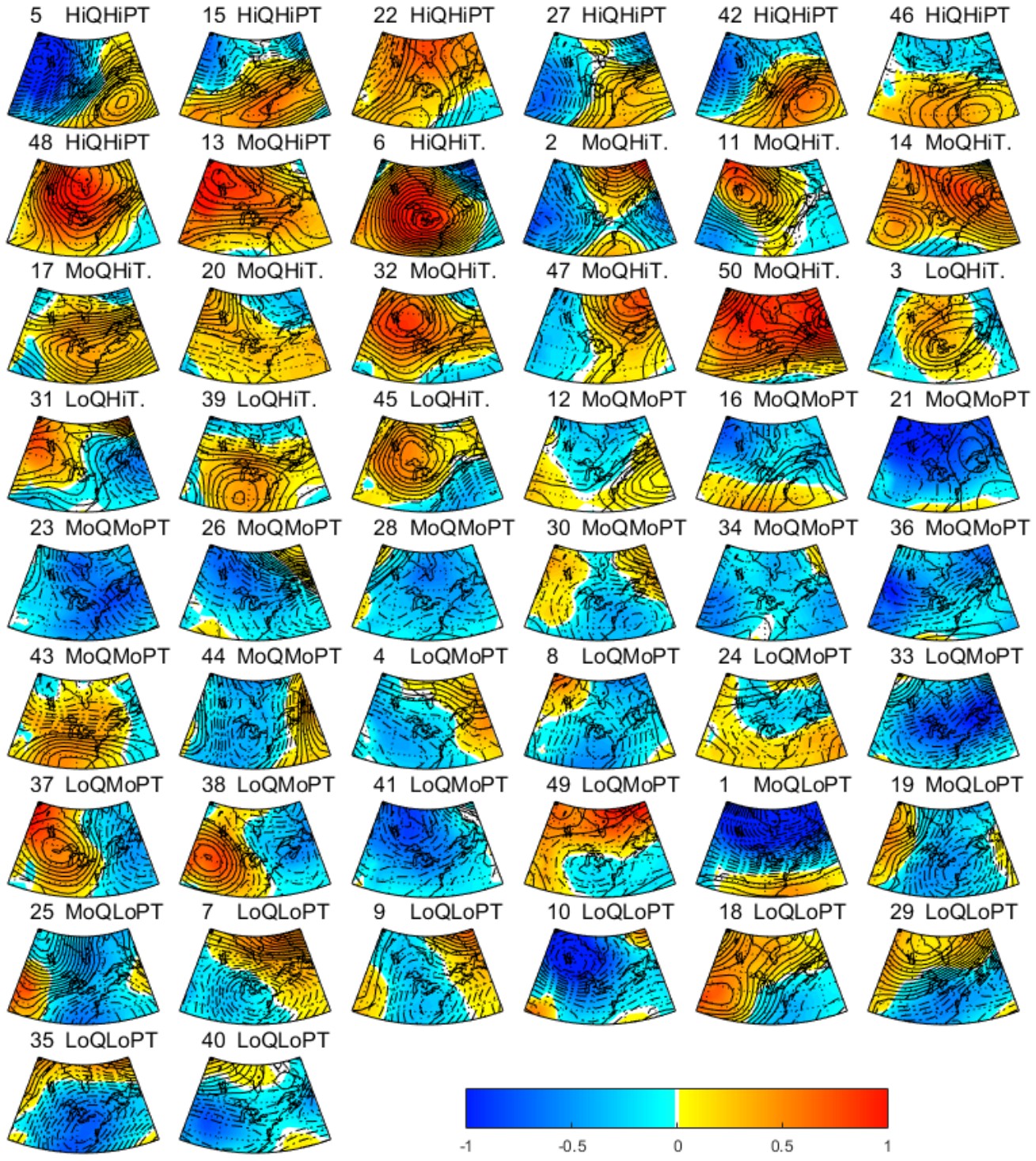

**Figure 11: Internal change of T850 (shade) and Z500 (black lines, interval 2m) between the historical and the 2040's period in January-February for each member.**

**Table 1: Geomorphic, land use, and soil characteristics of the four watersheds examined in this study**

| | Size (km²) | Altitude (m) | Land use (%) | | | | Soil type (%) | | |
|---|---|---|---|---|---|---|---|---|---|
| | | | Urban/Barren | Forest | Shrub | Crops/Grass | Sand | Loam | Clay |
| Big Creek | 571 | 179-336 | 1.9 | 17 | 0 | 81.1 | 78.6 | 6.4 | 15 |
| Grand River | 5091 | 178-531 | 7.1 | 11.9 | 0 | 80.9 | 30.4 | 31.6 | 38 |
| Thames River | 3061 | 215-423 | 6.9 | 5.4 | 0 | 87.7 | 14 | 46.7 | 39.4 |
| Credit River | 646 | 190-521 | 6.6 | 31.7 | 0 | 61.8 | 42.5 | 49.1 | 8.4 |

**Table 2 Parameter values after calibration (C: Calibrated, GIS: estimated by arcpy_GSFLOW, μ: GRUs average)**

| Parameter | Unit | Big Creek | Grand River | Thames River | Credit River | Spatial and temporal | Source |
|---|---|---|---|---|---|---|---|
| dday_intcp | Degrees days | -27 – -10 | -26 – -9 | -26 – -11 | -26 – -9 | monthly | C |
| dday_slope | Degrees days / °F | 0.38 – 0.41 | 0.38 – 0.42 | 0.38 – 0.42 | 0.38 – 0.42 | monthly | C |
| tmax_index | °F | 29.3 – 80 | 31.2 – 78 | 29.3 – 80 | 26.5 – 78.3 | monthly | C |
| jh_coef | per °F | 0.005 – 0.021 | 0.005 – 0.02 | 0.005 – 0.021 | 0.003 – 0.02 | monthly | C |
| Jh_coef_hru | per °F | 22 – 22.9 | 20.4 – 21.4 | 20.7 – 21.3 | 20.4 – 21.5 | GRU | GIS |
| Adjmix_rain | Decimal fraction | 0 | 0 | 1 | 0 | One | C |
| Cecn_coef | Calories per °C > 0 | 20 | 15 | 10 | 0 | One | C |
| emis_noppt | Decimal fraction | 0.757 | 0.757 | 0.757 | 0.757 | One | C |
| Fastcoef_lin | Fraction / day | 0.01 | 0.2 | 0.1 | 0.2 | One | C |
| Fastcoef_sq | none | 0.03 | 0.1 | 0.4 | 0.5 | One | C |
| Freeh2o_cap | inches | 0.07 | 0.01 | 0.01 | 0.01 | One | C |
| Gwflow_coef | Fraction / day | 0.05 | 0.05 | 0.06 | 0.03 | One | C |
| Potet_sublim | Decimal fraction | 0.1 | 0.75 | 0.1 | 0.6 | One | C |
| Smidx_coef | Decimal fraction | 0.02 | 0.05 | 0.04 | 0.001 | One | C |
| Smidx_exp | 1 / inch | 0.1 | 0.2 | 0.2 | 0.3 | One | C |
| Soil_rechr_max | inches | μ= 0.95 | μ= 0.87 | μ= 0.58 | μ= 2.8 | GRU | GIS+C |
| Soil_moist_max | inches | μ= 7.2 | μ= 2.9 | μ= 3 | μ= 3.1 | GRU | GIS |
| Tmax_allrain | °F | 34 | 35 | 33 | 36 | One | C |
| hru_percent_imperv | Decimal fraction | 0.1 – 0.6 | 0.1 – 0.6 | 0.1 – 0.6 | 0.1 – 0.6 | GRU | GIS |
| Ssr2gw_exp | none | 3 | 1 | 1.5 | 3 | One | C |
| Ssr2gw_rate | Fraction / day | μ= 0.23 | μ= 0.13 | μ= 0.12 | μ= 0.11 | GRU | GIS+C |
| Slowcoef_sq | none | μ= 0.79 | μ= 0.37 | μ= 0.21 | μ= 0.06 | GRU | GIS+C |
| Slowcoef_lin | Fraction / day | μ= 0.57 | μ= 0.008 | μ= 0.05 | μ= 0.02 | GRU | GIS+C |
| K_coef | hours | 2.8 – 8.4 | 1.6 – 3.2 | 1.78 – 3.56 | 1.35 – 2.68 | Segment | GIS+C |
| Pref_flow_den | Decimal Fraction | 0.1 | 0.1 | 0.1 | 0.2 | One | C |
| Rain_adj | Decimal Fraction | 0.77 – 0.86 | 0.69 – 1.12 | 0.92 – 1.04 | 0.87 – 0.94 | GRU Monthly | GIS |

| Snow_adj | Decimal Fraction | 0.96 – 1.06 | 0.69 – 1.12 | 0.92 – 1.04 | 0.72 – 0.76 | GRU Monthly | GIS |
|---|---|---|---|---|---|---|---|

**Table 3: Efficiency of PRMS model for best fit parameters**

| | Calibration | | Validation | |
|---|---|---|---|---|
| | NSE | PBIAS | NSE | PBIAS |
| Big Creek | 0.75 | 1.8 | 0.74 | 6.7 |
| Grand River | 0.71 | -5 | 0.69 | 1.7 |
| Thames River | 0.72 | -10.8 | 0.72 | -5.3 |
| Credit River | 0.71 | -0.1 | 0.65 | 18 |

**Table 4: Classes members, percentage of the ensemble in the class and average January-February percentage increase of streamflow between historical and 2040's period. The term in parenthesis indicates the standard deviation when the class has more than two members.**

| Name | Members | Percentage of the total ensemble | Q ( % increase) | | | |
|---|---|---|---|---|---|---|
| | | | Big Creek | Grand River | Thames River | Credit River |
| **HiQHiPT** | 5,15,22,27, 42,46,48 | 14% | 31.6 (8.1) | 48.3 (11.1) | 47 (9.6) | 53.7 (15) |
| **HiQHiT** | 6 | 2% | 24.9 | 44 | 39.4 | 46.8 |
| **MoQHiPT** | 13 | 2% | 24 | 35 | 35.7 | 34.1 |
| **MoQHiT** | 2,11,14,17, 20,32,47,50 | 16% | 21.2 (4.4) | 33.2 (3.2) | 32.2 (5.4) | 33.9 (2.8) |
| **MoQMoPT** | 12,16,21,23,26,28, 30,34,36,43,46 | 22% | 19.3 (5.2) | 33.7 (4.8) | 32.9 (6.1) | 34.1 (5.1) |
| **MoQLoPT** | 1,19,25 | 6% | 17.8 (1.5) | 31.2 (1.3) | 27.6 (2.5) | 32.7 (0.4) |
| **LoQHiT** | 3,31,39,45 | 8% | 10.6 (2.7) | 24.2 (1.7) | 23.5 (1.3) | 22 (3.4) |
| **LoQMoPT** | 4,8,24,33, 37, 38,41,49 | 16% | 13.8 (4) | 21.3 (4.1) | 22.7 (3.4) | 20.6 (7.3) |
| **LoQLoPT** | 7,9,10,18, 29,35,40 | 14% | 8.3 (7.8) | 18.8 (5.8) | 17.8 (6.4) | 18.6 (6.5) |