# Peer review of "Future shift in winter streamflow modulated by internal variability of climate in southern Ontario"

_Hydrology and Earth System Sciences, 2019_

## Referee Comment (RC1) · Anonymous Referee #1 · 5 Sep 2019

This paper looks at potential future shifts in climate and streamflow for four river catchments in southern Ontario. The CRCM5-LE RCP 8.5 scenario projections of air temperature and precipitation were used as input in the Precipitation Runoff Modelling System (PRMS) to determine future streamflows.

One conclusion of the work is the increase in winter streamflows in the future, particularly in the months of January and February. I find this very speculative because the bias between the observed and simulated flows for the historical period is greatest for these months. The bias is not adequately addressed in the paper and the uncertainties contributing to this bias are not adequately discussed. Hence, I recommend major

revisions be carried out before the paper is considered for publication.

Major comments include:

Page 1, Line 27: "glaciated or nival catchments" – why even mention this since southern Ontario is a region that has neither glaciated not nival areas?

Page 3, Line 20: You use the reference Marstrom et al. But I believe, PRMS was first developed by George Leavesley from USGS in the 1980s – shouldn't he be credited for the model development as well?

Page 5, Line 16: Please expound on the difference between observational and controlled streamflow.

Page 5, Line 16: Please explain the meaning of "controlled stream flow" and why CanGRD is used specifically to simulate it.

Page 5, Line 18: More discussion is required on the performance of the simulations of the historical period.

Page 5, Line 18: A comparison is required between historical and observed results to provide some confidence in the simulations.

Page 5, Lines 15 to 24: More discussion is required on model and data uncertainties, perhaps not here but elsewhere. Perhaps the bias correction is ok, but there may be some major issues with the hydrological model?

Figure 3: As stated above, the bias in flows for January and February are too large to be glanced over quickly and requires more attention in the paper, especially since you are making substantial conclusions from these periods with largest bias. Due to this major weakness in the paper, the rest of the paper loses credibility and the subsequent discussion seems moot.

Some editorial comments are:

Page 1, Line 28: period at end of sentence is missing

Page 2, Line 12: "conditions", not "condition"

Page 2, Line15: should read: . . . the GCMs'

Page 2, Line 22: closed bracket missing after Leduc et al., 2019)

Page 2, Line 27: should read: Seiller and Anctil (2014)

Page 2, Line 28: should read: Erler at al. (2018)

Page 3, Line 8: should read: . . . Brantford along (on?) the Grand River and London along (on?) the Thames River . . .

Page 3, Line 22: "The latter", not "These latest" - the former phrase refers to a position in sequence, the latter to a point in time.

Page 4, Line 25: reference should read: Ines and Hansen (2006)

Page 5, Line 11: reference should read: Deser et al (2014)

Page 5, Line 22: The simulated range . . . is "wide", not "high"? I'm referring to the second occurrence of "high" in that line.

I'll stop here. There are too many errors and I'll leave it to the editor to pick those up.

Comments to figures:

The color shading in the legend of Figure 3 is not consistent with the shading of the graphs. Also the color shading is not consistent with the legend shading in Figure 4. The graphs are very busy and hard to interpret, especially with the inconsistent shadings between graphs and legends. This needs to be fixed.

---

## Referee Comment (RC2) · Anonymous Referee #2 · 29 Sep 2019

General Comments I found the paper quite interesting and provides some substantial and important conclusions. Having said this, I think it really needs to be much more specific in the methodology, be clear on the assumptions that need to be and acknowledge a few fundamental issues with taking such an approach.

The model appears to have been calibrated for a reasonable period of time against what appears to be streamflow records. It is not clear where the streamflow records were obtained or where the locations of the gauges are. The authors should comment fifth cal/val statistics are sufficient for the analysis on climate change they propose. Also, the choice of Anuspline Homogenized (what they call CanGRD) data over per-

haps other data sets for forcing is not clear.

Consider the focus on snowmelt period and snowmelt simulations, the authors never discuss the appropriateness of the model physics for the snowmelt period. Does the PRMS model use an energy budget or temperature index. Is one method more appropriate for snowmlet, particularly in a climate change context, over another ? This should be at least mentioned.

Data used to derive the physiographic information to develop the model is not described, nor are the basins, except for very cursory comments. For example, there are many small control structures in these systems. The reader needs to know that and be made aware that they have or don't have an influence on the calibration or simulations.

The value of the paper appears to be in the messaging around the ensemble members results. Also, the attribution to synoptic patters provides some very interesting insights and the methodology seems reasonable, but the author would benefit from clearer explanations in sections 4.3. and 4..4. I find this very compelling and interesting, but it seems to get lost because the methodology confounds us in trying to understand what the authors are trying to do. I believe the intent and actual contribution of this work is important and should be published, but substantial clarification and structure to the manuscript is required.

Section 2.2. - comments Authors should state why they used PRMS instead of other models ? What is because it is computationally efficient ? has it been used by operational agencies in the region ? Some clarification is required. This section should include 2 parts. 1. model geo-fabric setup, including details around DEM and landcover (which ones) and how HRUs and routing is derived 2. forcing variables (what is necessary and how they are derived, where they come from) is not clear

Authors should describe better how the HRUs are generated. The reviewer presumes that a single dominant land type and soil type is used for each grid cell (as per the model documentation for PRMS). Authors should define how the grid (which are the

same as HRUs ?) are defined in this application of the HRU, and specify that each grid is treated as an HRU. PRMS also requires stream networks, sub-basins, lakes to be defined. a few lines around how this was done or perhaps a schematic on how PRMS was implemented here would be worthwhile. Perhaps a figure similar to Figure 4 in the PRMS user manual but for the author's Big Creek application would be useful. It is difficult to get a sense of how the model was setup for this application.

The last part of section 2.2. describing the meteorological forcing used is also quite confusing. CanGRD (according the Environment Canada) is a monthly, seasonal and annual product. Perhaps the author is referring to the homogenized data used in the development of of the product produced by McKinley, which based on the article cited which I read, does not have a formal name. Also, there are a lot of other products available, so some justification as to what this product, which is quite a bit older thanks some of the more recent published data such as WATCH or CAPA, is being used. Also, can you clarify which streamflow gauges were used ?

Lastly, you mention muskingum routing, but it is not completely clear how this was calibrated. This is likely the most sensitive parameter the the NS criteria. Can you confirm how sensitive the results were to the routing ?

Section 2.3 - comments. A more complete description of the data developed in CanRCM-LE would be useful. I was required to lookup what this data set contained and how the ensembles were generated. I think the authors should actually include some level of detail here.

Section 2.4 - Comments This section is extremely unclear. I would recommend the authors describe what AHC is and at minimum make some reference to how the various ensembles were classified. What is the purpose of doing the ACH analysis, and is there a reference ?

Section 3.1 The methodology becomes clearer after reading this section. I would encourage the authors to maybe re-write some of sections 2.2 to clarify the approach.

It seems that what was done was 1. Calibrate these basins for use with PRMS using historic homogenized and gridded daily (5 years) data. 2. Using the CRCM-LE historic biased corrected forcing for the simulations and run ensembles. The authors should perhaps take a bit of time to describe why this approach was taken e.g. why not calibrate to a 10 year period. Are there any concerns about perhaps parameters values changing under a different climate regime ? Are you concerned about calibrating with Anuspline but driving the model with a different precipitation model, even if it was bias corrected. Some commentary here is necessary. The authors looked at ET, and I assume it was from the PRMS model. Why not use RCM or at least see what the RCM produces ? Since it is based on CLASS, should dit not be a bit more realistic than PRMS ?

The authors show in figure 5 increases in temp and precipitation. Can you clarify if this is the bias corrected values or original CRCM5-LE.

Section 3.2 A paragraph describing what ACH with a reference is required either in the methodology or here. Up to this point in the text, it is unclear why the ACH approach is even necessary. It does get clarified, but should be referenced and explained in section 2. The division between hi-lo and moderate and conglomeration of weather and flow classes seems a bit subjective. The authors should be clearer on how they chose to group these. It is not clear how you have a HiT category since P and T are combined. One assumes that the change in P is simply small. Also the whole section is difficult to follow and essentially describes what is in the table and on the plots, but it doesn't really tell me what I think it is trying to tell me. It seems that this is al about attribution of the change in flows. Is it caused by increases in T, P, or both. Section 3.2 does not really assist me in understanding.

Section 4.1 The authors never mention issues around frozen soils, freeze that cycles or river ice formation. River ice can have a large influence on hydrometric measurements and rating curves. Often it is too dangerous to take flow measurements in the winter so many flow values are estimated that time of year. The authors need to acknowledge

something on uncertainty in winter measurements.

Section 4.2 and 4.3 The synoptic discussions are interesting but a bit confusing. This really need to be better explained and expanded.

Specific comments Page 2- Line 25-30 - Did you mean just limited members from CRCM5-LE or a different ensemble from Seiller and Anctil ? Same for Erler ? It would be useful if you clarified if you are using these new ensembles for the first time or you are the first to use all 50 as other authors had only used select ensemble members grin the same set. This is a bit ambiguous. Page 2 Line 30. For readability, it would be useful to add a sentence here as to why using 50 ensemble is important. Page 3: - line 22 should use "computational time" or "model computation time " instead of model time. Page 3 - Line 25-27 - The authors should expand this to either include the equation or explain this better. The reader who is not completely familiar with PRMS will not understand what the coefficients are used for or what they mean. Page 4 - line 9 - please indicate the time step.

---

## Author Comment (AC1) · 26 Oct 2019

**We would like to thank the reviewer for their comments on our paper. Please find our answers below:**

**Reviewer #1:**

This paper looks at potential future shifts in climate and streamflow for four river catchments in southern Ontario. The CRCM5-LE RCP 8.5 scenario projections of air temperature and precipitation were used as input in the Precipitation Runoff Modelling System (PRMS) to determine future streamflows. One conclusion of the work is the increase in winter streamflows in the future, particularly in the months of January and February. I find this very speculative because the bias between the observed and simulated flows for the historical period is greatest for these months. The bias is not adequately addressed in the paper and the uncertainties contributing to this bias are not adequately discussed. Hence, I recommend major revisions be carried out before the paper is considered for publication.

**Please find below our answer to address the bias.**

Major comments include:

Page 1, Line 27: "glaciated or nival catchments" – why even mention this since southern Ontario is a region that has neither glaciated not nival areas?

**"Glaciated and nival catchment" will be replaced by ''snow-dominated region'' in the new version of the manuscript. Snow is a very important component of the hydrology in southern Ontario and we found it important to mention that similar shifts in streamflow were observed in other snow-dominated catchments around the world.**

Page 3, Line 20: You use the reference Marstrom et al. But I believe, PRMS was first developed by George Leavesley from USGS in the 1980s – shouldn't he be credited for the model development as well?

**The reference Leavesley et al., (1983) will be added to the manuscript.**

Page 5, Line 16: Please expound on the difference between observational and controlled streamflow.

**Observational streamflow is the streamflow measured at each watershed outlet and controlled streamflow is the streamflow simulated by PRMS using observed temperature and precipitation. These details will be added to the manuscript.**

Page 5, Line 16: Please explain the meaning of "controlled stream flow" and why CanGRD is used specifically to simulate it.

**Controlled streamflow is the streamflow simulated by PRMS using observed temperature and precipitation. We called it control to not confuse it with the streamflow simulated using biased corrected CRCM5-LE temperature and precipitation (HIST). It also needs to be distinguished from the streamflow measured at the outlet (OBS). The expression "controlled streamflow" will be removed to avoid confusion and details about OBS and CTL will be added.**

**CanGRD meteorological dataset was used in a previous study focusing in southern Ontario (Wazneh et al., 2017). This dataset is often referred to as NRCANmet in number of other studies and is the most commonly used gridded climate dataset in Canada (Werner et al., 2019). The dataset was produced using station-based observations from Environment Canada and Natural Resources Canada and the gridding was accomplished using the Australian National University Spline (ANUSPLIN) with latitude, longitude and elevation as predictors (Hutchinson et al., 2009). To avoid confusion with a monthly product created by Environment Canada called CanGRD, the dataset will be renamed NRCANmet in the entire manuscript.**

Page 5, Line 18: More discussion is required on the performance of the simulations of the historical period.

Page 5, Line 18: A comparison is required between historical and observed results to provide some confidence in the simulations.

Page 5, Lines 15 to 24: More discussion is required on model and data uncertainties, perhaps not here but elsewhere. Perhaps the bias correction is ok, but there may be some major issues with the hydrological model?

Figure 3: As stated above, the bias in flows for January and February are too large to be glanced over quickly and requires more attention in the paper, especially since you are making substantial conclusions from these periods with largest bias. Due to this major weakness in the paper, the rest of the paper loses credibility and the subsequent discussion seems moot.

**A paragraph discussing the historical discrepancy between OBS, CTL and HIST was included in the discussion of the submitted manuscript (Section 4.1). The streamflow from CTL is clearly overestimated by PRMS in Big Creek and Thames River as compared to OBS but the annual cycle was well reproduced by PRMS. PRMS have been previously used for these watersheds and snow processes in Big Creek watershed were well simulated (Champagne et al., 2019). Overestimation of streamflow may be from the ANNUSPLIN method that overestimates precipitation in this region (Newlands et al., 2011). Despite the biases from ANNUSPLIN, NRCANmet is the most widely used gridded dataset in Canada (Werner et al., 2019) and can be used with confidence. Further discussion on overestimation from ANNUSPLIN will be added to the manuscript. The authors are also aware that the results are from a single model chain and it will be relevant in the future to use other models. We will therefore mention this concern in the conclusion of the manuscript: ''Despite a large number of regional climate simulations used here to drive a hydrological model, the 50 member ensemble used here represents internal variability derived from a single model chain (CanESM2, CRCM5 and PRMS). As a result, this ensemble does not consider other important sources of uncertainty from emission scenario and model structure''.**

Some editorial comments are:

Page 1, Line 28: period at end of sentence is missing

Page 2, Line 12: "conditions", not "condition"

Page 2, Line15: should read: . . . the GCMs'

Page 2, Line 22: closed bracket missing after Leduc et al., 2019)

Page 2, Line 27: should read: Seiller and Anctil (2014)

Page 2, Line 28: should read: Erler at al. (2018)

Page 3, Line 8: should read: . . . Brantford along (on?) the Grand River and London along (on?) the Thames River . . .

Page 3, Line 22: "The latter", not "These latest" - the former phrase refers to a position in sequence, the latter to a point in time.

Page 4, Line 25: reference should read: Ines and Hansen (2006)

Page 5, Line 11: reference should read: Deser et al (2014)

Page 5, Line 22: The simulated range . . . is "wide", not "high"? I'm referring to the second occurrence of "high" in that line.

**These errors will be corrected**

I'll stop here. There are too many errors and I'll leave it to the editor to pick those up.

**The grammar and typographic errors will be corrected in the entire manuscript**

Comments to figures: The color shading in the legend of Figure 3 is not consistent with the shading of the graphs. Also the color shading is not consistent with the legend shading in Figure 4. The graphs are very busy and hard to interpret, especially with the inconsistent shadings between graphs and legends. This needs to be fixed.

**The shade colors in the legend will be modified to correspond to colors from the graphs. To make the graph less busy we will remove the horizon 2080s which were not included in the analyses (Figure R1). Figures 4 and 5 will be similarly modified.**

**References:**

Champagne, O., Arain, M. A. and Coulibaly, P.: Atmospheric circulation amplifies shift of winter streamflow in Southern Ontario, Journal of Hydrology, 124051, doi:10.1016/j.jhydrol.2019.124051, 2019.

Leavesley, G. H., Lichty, R. W., Troutman, B. M. and Saindon, L. G.: Precipitation-runoff modeling system; user's manual., 1983.

Newlands, N. K., Davidson, A., Howard, A. and Hill, H.: Validation and inter-comparison of three methodologies for interpolating daily precipitation and temperature across Canada, Environmetrics, 22(2), 205–223, doi:10.1002/env.1044, 2011.

Werner, A. T., Schnorbus, M. A., Shrestha, R. R., Cannon, A. J., Zwiers, F. W., Dayon, G. and Anslow, F.:
A long-term, temporally consistent, gridded daily meteorological dataset for northwestern North
America, Scientific Data, 6(1), doi:10.1038/sdata.2018.299, 2019.

---

## Author Comment (AC2) · 26 Oct 2019

**We would like to thank the reviewer for their comments on our paper. Please find our answers below:**

**Reviewer #2:**

General Comments I found the paper quite interesting and provides some substantial and important conclusions. Having said this, I think it really needs to be much more specific in the methodology, be clear on the assumptions that need to be and acknowledge a few fundamental issues with taking such an approach.

The model appears to have been calibrated for a reasonable period of time against what appears to be streamflow records. It is not clear where the streamflow records were obtained or where the locations of the gauges are. The authors should comment fifth cal/val statistics are sufficient for the analysis on climate change they propose. Also, the choice of Anuspline Homogenized (what they call CanGRD) data over perhaps other data sets for forcing is not clear.

**The streamflow record was taken from Water Survey Canada and the gauges are located at the outlet of each watershed (Figure 1 in the main manuscript). Clarifications for these gauges and a reference to Figure 1 will be added to the manuscript.**

**As stated at the end of section 2.2, NSE and PBIAS were satisfactory (Moriasi et al., 2007). As shown in our previous study (Champagne et al., 2019), winter streamflow and snow processes were also satisfactorily simulated. A reference to Champagne et al., (2019) will be added in this section 2.2.**

**CanGRD meteorological dataset was used in a previous study focusing in southern Ontario (Wazneh et al., 2017). This dataset is often referred to as NRCANmet in number of other studies and is the most commonly used gridded climate dataset in Canada (Werner et al., 2019). Justification for the choice of this dataset will be added in the manuscript.**

Consider the focus on snowmelt period and snowmelt simulations, the authors never discuss the appropriateness of the model physics for the snowmelt period. Does the PRMS model use an energy budget or temperature index. Is one method more appropriate for snowmlet, particularly in a climate change context, over another ? This should be at least mentioned.

**The ability of PRMS to simulate snow processes will be added in the manuscript with a reference to Champagne et al. (2019). The Snowmelt algorithm uses an energy balance approach based on temperature and precipitation data. The advantage of this method is that snowmelt is better conceptualized than a temperature index approach and does not use data projections that may be difficult to obtain (e.g. radiation).**

Data used to derive the physiographic information to develop the model is not described, nor are the basins, except for very cursory comments. For example, there are many small control structures in these systems. The reader needs to know that and be made aware that they have or don't have an influence on the calibration or simulations.

**The data used to derive physiographic information are High Resolution Digital Elevation Model (HRDEM) and the Canadian Land Cover CIRCA 2000, both furnished by Natural**

**Resources Canada, and the surficial geology of southern Ontario furnished by The Ontario Ministry of Northern Development, Mines and Forestry. These data sources will be added to the new manuscript.**

**The control structures were not taken into consideration in the model set up. The authors are aware that these structures can play a role in the modulation of streamflow. However, our study investigates the change in average streamflow, while control dams have greater impact on specific peak flows. The dams have very limited impact on the average streamflow calculated over a 30-year period.**

The value of the paper appears to be in the messaging around the ensemble members results. Also, the attribution to synoptic patters provides some very interesting insights and the methodology seems reasonable, but the author would benefit from clearer explanations in sections 4.3. and 4..4. I find this very compelling and interesting, but it seems to get lost because the methodology confounds us in trying to understand what the authors are trying to do. I believe the intent and actual contribution of this work is important and should be published, but substantial clarification and structure to the manuscript is required.

**Clarification of the methodology will be addressed according to the comments below.**

Section 2.2. - comments Authors should state why they used PRMS instead of other models ? What is because it is computationally efficient ? has it been used by operational agencies in the region ? Some clarification is required. This section should include 2 parts. 1. model geo-fabric setup, including details around DEM and landcover (which ones) and how HRUs and routing is derived 2. forcing variables (what is necessary and how they are derived, where they come from) is not clear

**PRMS was used in this study because has been satisfactorily used in other snow dominated regions and was already applied for these same watersheds (Champagne et al., 2019). According to this study, PRMS reconstructed snow processes in Big Creek watershed. A Few sentences explaining the choice of PRMS will be added in section 2.1.**

**The model setup was done using Arcpy-GSFLOW as described in Gardner et al., (2011). The PRMS modules used in these watersheds have been described in Champagne et al., (2019). These two references will be added to the manuscript. We will also add more information on the datasets used for the setup (described above).**

**The forcing variables are minimum and maximum temperature and precipitation at 10km spatial resolution using the NRCANmet dataset. A short explanation of the dataset and references will be added to the manuscript.**

Authors should describe better how the HRUs are generated. The reviewer presumes that a single dominant land type and soil type is used for each grid cell (as per the model documentation for PRMS). Authors should define how the grid (which are the same as HRUs ?) are defined in this application of the HRU, and specify that each grid is treated as an HRU. PRMS also requires stream networks, sub-basins, lakes to be defined. a few lines around how this was done or perhaps a schematic on how PRMS was implemented here would be worthwhile. Perhaps a

figure similar to Figure 4 in the PRMS user manual but for the author's Big Creek application would be useful. It is difficult to get a sense of how the model was setup for this application.

**HRUs consisted of surface grid cells of 200 m² for Big Creek and Credit River watersheds and 400 m² for Grand River and Thames River. For each HRU, the percentage of each land use type (bare soil, grass, shrubs, coniferous trees and deciduous trees) and soil type (sand, loam or clay) was calculated by Arcpy-GSFLOW. Some parameters were estimated using these percentages while other PRMS calculations were based on an integer number corresponding to the most dominant land use or soil type. These precisions will be added to the manuscript.**

**The stream network was computed with ARCGIS using DEM and accumulation threshold was determined empirically to make the conceptual stream network match the stream positions from satellites maps. Since only one hydrometric station was used for each watershed, these watersheds were considered as one sub-basin. HRU's are considered as lakes when the entire area is covered by water as determined by land use.**

**The description of PRMS setup will be greatly improved in the new manuscript. Given that HRUs are grids of similar sizes, we judged it was not necessary to include a figure similar to figure 4 from the PRMS manual.**

The last part of section 2.2. describing the meteorological forcing used is also quite confusing. CanGRD (according the Environment Canada) is a monthly, seasonal and annual product. Perhaps the author is referring to the homogenized data used in the development of of the product produced by McKinley, which based on the article cited which I read, does not have a formal name. Also, there are a lot of other products available, so some justification as to what this product, which is quite a bit older thanks some of the more recent published data such as WATCH or CAPA, is being used. Also, can you clarify which streamflow gauges were used ?

**As previously stated, CanGRD referred to as NRCANmet in number of studies and is the most commonly used gridded climate dataset in Canada (Werner et al., 2019). This dataset was produced using station-based observations from Environment Canada and Natural Resources Canada and the gridding was accomplished using the Australian National University Spline (ANUSPLIN) with latitude, longitude and elevation as predictors (Hutchinson et al., 2009). The dataset will be renamed NRCANmet in the entire manuscript.**

Lastly, you mention muskingum routing, but it is not completely clear how this was calibrated. This is likely the most sensitive parameter the the NS criteria. Can you confirm how sensitive the results were to the routing ?

**The muskingum routing was calibrated by fitting the Muskingum storage coefficient (K_coef) using the Normal Root Mean Squared Error (NRMSE) between daily and monthly observed and simulated streamflow. The inter-segment variability of K_coef was estimated using the segment length and the slope. This variability was preserved during the**

**calibration by multiplying K_coef of each segment by the same coefficient. The results were sensitive to the routine and especially the timing of the streamflow and the amplitude of high flows.**

Section 2.3 - comments. A more complete description of the data developed in CanRCM-LE would be useful. I was required to lookup what this data set contained and how the ensembles were generated. I think the authors should actually include some level of detail here.

**Details and references on the development of CRCM5-LE will be added in the manuscript.**

Section 2.4 - Comments This section is extremely unclear. I would recommend the authors describe what AHC is and at minimum make some reference to how the various ensembles were classified. What is the purpose of doing the ACH analysis, and is there a reference ?

**The AHC will be described in the new manuscript as follows: To classify n members, the Euclidean distance between each pair of members are first calculated. The two members with the smallest Euclidean distance are merged into a single class. Then, the Euclidean distance between this class of members and the n-2 other members is calculated. The two members or classes of members with the smallest Euclidean distance are merged and a new Euclidean distance is calculated. This process is repeated n-1 times in total, until all classes of members have been merged into a single class. The dendrogram in Figure 6 shows the successive merging from the first merging using all members (bottom) to the last merging creating a single class (top) (Figure 6). Each merging is associated with a variance of Euclidean distance between the two successive merged classes (Y axis). The highest variance difference between two successive merging classes show what is the number of classes corresponding to the highest interclasses variance.**

Section 3.1 The methodology becomes clearer after reading this section. I would encourage the authors to maybe re-write some of sections 2.2 to clarify the approach. It seems that what was done was 1. Calibrate these basins for use with PRMS using historic homogenized and gridded daily (5 years) data. 2. Using the CRCM-LE historic biased corrected forcing for the simulations and run ensembles. The authors should perhaps take a bit of time to describe why this approach was taken e.g. why not calibrate to a 10 year period. Are there any concerns about perhaps parameters values changing under a different climate regime ? Are you concerned about calibrating with Anuspline but driving the model with a different precipitation model, even if it was bias corrected. Some commentary here is necessary. The authors looked at ET, and I assume it was from the PRMS model. Why not use RCM or at least see what the RCM produces ? Since it is based on CLASS, should dit not be a bit more realistic than PRMS ?

**The part 2.2 will be clarified in the new manuscript to better explain the dataset used to calibrate the model and the dataset used for the future projections. As stated in the manuscript, the calibration period was 20 years (1989-2008), not 5. The reviewer refers probably to the warm-up period (1984-1989) or the validation period (2009-2013).**

**The authors were concerned about calibrating with ANUSPLIN and driving the model with CRCM5-LE. This is why climate observations and historical data from CRCM5-LE**

were compared and streamflow computed with ANUSPLIN and CRCM5-LE were compared as well in the historical period. Section 4.1 described the discrepancy between simulations using ANUSPLIN and CRCM5-LE in the historical period.

**We appreciate the suggestion to compare ET from PRMS and RCM. However, due to the size of the dataset, the extraction and transfer of the variables are very time consuming and only temperature and precipitation were extracted. We will suggest using ET from CRCM5-LE in future studies in the new version of the manuscript.**

The authors show in figure 5 increases in temp and precipitation. Can you clarify if this is the bias corrected values or original CRCM5-LE.

**Temperature and precipitation shown in Figure 5 are the bias corrected values. This information will be added into the new manuscript.**

Section 3.2 A paragraph describing what ACH with a reference is required either in the methodology or here. Up to this point in the text, it is unclear why the ACH approach is even necessary. It does get clarified, but should be referenced and explained in section 2. The division between hi-lo and moderate and conglomeration of weather and flow classes seems a bit subjective. The authors should be clearer on how they chose to group these. It is not clear how you have a HiT category since P and T are combined. One assumes that the change in P is simply small. Also the whole section is difficult to follow and essentially describes what is in the table and on the plots, but it doesn't really tell me what I think it is trying to tell me. It seems that this is al about attribution of the change in flows. Is it caused by increases in T, P, or both. Section 3.2 does not really assist me in understanding.

**AHC will be clarified in section 2.4 as stated above. The AHC was first used to group the members into classes of similar change of streamflow. The AHC uses the Euclidean distance between members and it can be applied simultaneously using different variables (here the variables are the evolution of streamflow for each of the 4 watersheds). The AHC was applied to the standardized change of streamflow to avoid the Euclidean distance being dependent on large changes in one watershed. The AHC constructs these classes by maximizing the interclass variance. Therefore the classes are not arbitrary. The division between Hi, Low and moderate is based on the results of the AHC. Three classes are the most pertinent choice to maximize the interclass variance of streamflow change (Shown in figure 6). The variance between classes is maximal when the vertical distance between 2 successive merging is maximal. The labels Hi indicated the highest increase in streamflow while Lo indicated the lowest increase. Moderate is the class in between. High, Low and moderate are relative to other members and do not refer to an absolute high or low increase in streamflow.**

**The AHC was also applied to the standardized change of the two variables, temperature and precipitation (Figure 6 diagram at the bottom). The objective was to group the members so we did not have a member isolated in a single class. If we remove member #33 it is clear that the number of classes with the lowest interclass variance is 4.**

The conglomeration of streamflow and weather classes is not subjective because it is simply done by splitting the 3 streamflow classes into weather subclasses (e.g. members with simultaneously High Q and High PT in the same class, High Q and moderate PT in another class…etc…). The reviewer is referred to the Table 4 in the main manuscript depicting the streamflow and weather classes labels.

Figure 6 (right plot) shows that the HiT grouped member with high temperature change but not high precipitation change (Orange circle). For concision we decided to call it HiT. The right panel of figure 6 shows that the construction of the weather classes was not subjective but are formed from members that are similar in term of both precipitation and temperature change.

Explanations for the causes of streamflow change was described in the discussion part. Part 3.2 will be modified to avoid repetitions from the graphs.

Section 4.1 The authors never mention issues around frozen soils, freeze that cycles or river ice formation. River ice can have a large influence on hydrometric measurements and rating curves. Often it is too dangerous to take flow measurements in the winter so many flow values are estimated that time of year. The authors need to acknowledge something on uncertainty in winter measurements.

We simulated streamflow during frozen and not frozen soil conditions in Big Creek watershed and the difference was not significant (Figure R2). We used a lag of three days between the conditions of the soil and the streamflow because rain and/or snowmelt events take 3 days to form a peak at the outlet. We also tested lags of 1 to 6 days (6 days given the best correlation between seasonal average temperature/precipitation and seasonal average streamflow) and the results were not significantly different. We can therefore conclude that frozen ground does not have a significant impact on streamflow.

River ice can have an impact in gauge measurements, and this will be acknowledged in the new version of the manuscript.

Section 4.2 and 4.3 The synoptic discussions are interesting but a bit confusing. This really need to be better explained and expanded.

These sections will be rewrite as follow:

4.3 Consistency in the weather classes

The weather classes are associated to specific changes in atmospheric conditions (Figure 9) but are composed from an average of members that have their own signature. Change in Z500 anomalies for each member are depicted in Figure 11 to investigate the variability between members. The members that comprise classes HiPT show high Z500 anomalies enhance in the east coast consistently for six members while for two members (#13 and #48) the high increase in Z500 anomalies is centered north from the Great Lakes. Eight members of the class LoPT show strong Z500 decrease in the east coast but in two members (#1 and #10) the decline is centered in the northern side of the Great Lakes. HiT shows generally

Z500 increase centered on the Great Lakes but four of the thirteen members depict a different pattern (#2, #20, #31 and #47). Finally, members from MoPT show generally a decrease in Z500 but we observe a high diversity in the change of circulation patterns. Members from MoPT depict a lower Z500 gradient compared to other classes suggesting a lower contribution of internal variability of climate to the total change in atmospheric conditions for this class. Despite the atmospheric anomalies differences between members that predict similar local weather conditions, this study gives a good probabilistic overview on how the change in regional atmospheric anomalies will impact local weather.

**4.4 Lag between local climate conditions and streamflow**

The results of this study show that interclass variability in the increase of streamflow is mostly driven by temperature and precipitation variability in January-February. The members with the highest increase in precipitation and temperature (HiPT) are the members associated with the highest streamflow increases and the members associated with the lowest increase in precipitation and temperature (LoPT) show the lowest streamflow increase (LoQLoPT) (Table 4). Other LoPT members show comparatively higher streamflow (MoQLoPT) but this result can be explained by more precipitation and snowfall despite a lower warming (Figure 7).

Within the other two weather classes, HiT and MoPT, a similar change in January-February weather conditions translates to a large range in streamflow projections (Table 4). These discrepancies between the evolution of weather conditions and streamflow volume in January-February can be associated to a delay between weather conditions change and streamflow change. To account for the routing delay between rain/snowmelt events and streamflow observed at the outlet, our analyses use a lag-time of 6 days between the precipitation/temperatures and the streamflow but a delay between weather conditions and streamflow may occur due to remaining snowpack from December. This hypothesis is invalidated by the low variability among MoPT and HiT members in term of change in late December snowpack (Figure 10). The delay between weather conditions and streamflow can also be due to groundwater recharge/discharge variability. The lower streamflow increase in LoQHiT is associated simultaneously with a lower increase in groundwater flow and a lower increase in November-December precipitation amount (Figure 10). To confirm the connexion between fall precipitation and winter groundwater flow we calculated the inter-members correlation between November-December change in precipitation amount and the January-February change in groundwater flow and we found a coefficient of correlation close to 0.7. This result show that winter streamflow can be modulated by atmospheric conditions occurring as early as the fall season.

Another explanation for the discrepancy between change in weather conditions and change in streamflow may be the timing in precipitation and temperature change. Our results show that snowpack remaining at the end of January-February is decreasing at a higher rate for MoQMoPT members as compared to LoQMoPT members and for MoQHiT members compared to LoQHiT members (Figure 10). This result is likely due to more snowmelt

**simulated in the MoQ members (Figure 7). In the same time, the evolution of precipitation and temperature are similar between LoQMoPT and MoQMoPT and between MoQHiT and LoQHiT (Figure 7). These results show that to explain changes in snow processes the average change in precipitation and temperature is not sufficient and their timing must be taken into consideration. An increase in precipitation simultaneously with an increase in temperature is likely to produce rain with a direct impact on streamflow. On the contrary, when increase in temperature is mostly happening in January while February is affected by wetter and cooler conditions, snowpack is likely to contribute to the streamflow later in spring.**

**These discussions emphasize the need to study the succession of different atmospheric patterns that occur weeks or months before a discharge event.**

Specific comments

Page 2- Line 25-30 - Did you mean just limited members from CRCM5-LE or a different ensemble from Seiller and Anctil ? Same for Erler ? It would be useful if you clarified if you are using these new ensembles for the first time or you are the first to use all 50 as other authors had only used select ensemble members grin the same set. This is a bit ambiguous.

**These studies used other ensembles that have only 4 or 5 members. CRCM5-LE was not used before in north-eastern North America as input in hydrological models. This sentence will be modified for clarity.**

Page 2 Line 30. For readability, it would be useful to add a sentence here as to why using 50 ensemble is important.

**We will add a sentence explaining that 50 members are important because it depicts a large range of internal variability of climate and is appropriate for a probabilistic approach.**

Page 3: - line 22 should use "computational time" or "model computation time " instead of model time.

**We will change this part to read ''reduce the parametrization computation time''.**

Page 3 - Line 25-27 - The authors should expand this to either include the equation or explain this better. The reader who is not completely familiar with PRMS will not understand what the coefficients are used for or what they mean.

**This part will be clarified with reference to a previous paper that used PRMS in these watersheds (Champagne et al., 2019). The reader will be referred to Markstorm for details on PRMS that are not fundamental for understanding the manuscript and are common to all watersheds using PRMS.**

Page 4 - line 9 - please indicate the time step.

**The timestep (daily) will be added.**

**References:**

Champagne, O., Arain, M. A. and Coulibaly, P.: Atmospheric circulation amplifies shift of winter streamflow in Southern Ontario, Journal of Hydrology, 124051, doi:10.1016/j.jhydrol.2019.124051, 2019.

Gardner, A. S., Moholdt, G., Wouters, B., Wolken, G. J., Burgess, D. O., Sharp, M. J., Cogley, J. G., Braun, C. and Labine, C.: Sharply increased mass loss from glaciers and ice caps in the Canadian Arctic Archipelago, Nature, 473(7347), 357–360, doi:10.1038/nature10089, 2011.

Leavesley, G. H., Lichty, R. W., Troutman, B. M. and Saindon, L. G.: Precipitation-runoff modeling system; user's manual., 1983.

Moriasi, D. N., Arnold, J. G., Van Liew, M. W., Bingner, R. L., Harmel, R. D. and Veith, T. L.: Model evaluation guidelines for systematic quantification of accuracy in watershed simulations, Transactions of the ASABE, 50(3), 885–900, 2007.

Newlands, N. K., Davidson, A., Howard, A. and Hill, H.: Validation and inter-comparison of three methodologies for interpolating daily precipitation and temperature across Canada, Environmetrics, 22(2), 205–223, doi:10.1002/env.1044, 2011.

Werner, A. T., Schnorbus, M. A., Shrestha, R. R., Cannon, A. J., Zwiers, F. W., Dayon, G. and Anslow, F.: A long-term, temporally consistent, gridded daily meteorological dataset for northwestern North America, Scientific Data, 6(1), doi:10.1038/sdata.2018.299, 2019.

---

## Editor Decision (ED1)

Dear authors,

Thank you for your responses to the reviewer comments and for the revision of the manuscript. Both reviewers were positive about the manuscript and recommended to publish it after some considerable revisions. The results showing increases in winter discharge in key watersheds within southern Ontario under future climate are of interest, and the study helps improve understanding of the potential hydrological impacts. However, the revised manuscript suffers from a number of issues of clarity and other problems, and will require further revisions to bring it up to the quality standards for this journal. Both reviewers offered constructive and helpful comments on how this paper needs to be improved, but I find these have not yet been fully addressed. A more careful and thorough effort is required.

Major Issues

1. The model must be more fully explained and justified as to its appropriateness for use in this region and under changing climates. Why is it appropriate for use here and what are its main limitations? Simply referring to the fact that others have used it and citing your earlier paper are not sufficient. The snowmelt routine appears to be a simple temperature index approach (i.e. a snow energy balance approach explicitly accounts for turbulent and radiative energy exchanges), and here the only inputs are temperature and precipitation. So why and how can this be justified under future climates?

2. There needs to be more detail and explanation of the model setup, parameterization, and calibration/validation. Both reviewers were adamant about this. More discussion is needed on model and data uncertainties, especially given the bias in simulated winter flows as noted by reviewer #1. The comments by reviewer #2 included a number of important issues to address regarding model setup and geofabric. Why is it ok to neglect control structures and reservoirs, especially on the Grand River, where there are a number of flood control dams? (See specific comments further below.)

3. The ascending hierarchical classification needs to be better described, and perhaps better illustrated, as it remains quite unclear. The reader needs to understand this. Why focus on runoff response groupings, given the non-linear nature of runoff, as opposed to strictly synoptic climatological patterns?

Detailed Comments

I refer to page and line numbers for the "clean" (i.e. non marked-up) version of the ms. My comments here are meant to help specifically address some of the reviewer concerns and to flag other issues that need to be dealt with.

P1, L12 and throughout: what is "internal variability of climate"? This term is central to the paper, but it isn't made entirely clear what this actual means. Does it refer to variation among the ensemble of climate model outputs?

P1, L18-22: the short summary of results is very unclear. What is meant by the terms in parentheses? The reader can't understand this by just reading the abstract. How significant is it that 14% of the

ensemble members predict a high increase?  What about the rest of the ensemble?  More importantly, what is the magnitude and variation of projected flow changes?

P1, L22: what does the 16% refer to?  This isn't clear.

P1, L22: what is "internal variability of hydrological projections"?

P1, L30: the "choices" are really cascading sources of uncertainty throughout the modelling process. And does this link in to the concept of internal variability of climate?  There is an opportunity to explain all of this more clearly here.

P2, L15: "future climate data should not be used…" – do you mean that coarse-scale climate model outputs shouldn't be used directly?

P2, L29: does a large ensemble assess the entire range of internal variability?  Does this not also depend on selection of RCP, GCM, downscaling method, bias correction, hydrological model, parameterization, etc.?

P2, L30: instead of "processes", do you mean "responses"?

Introduction section: In general, this section should be a bit more clear on the overall purpose and objectives of this study, and on how it builds on previous work to advance understanding.  What is new and how and why is it important?

P3, L8-9: What about other urban areas such as Kitchener–Waterloo and others?

Section 2.1: This section should describe the major landcover types in more detail, and a bit on the climate and the hydrological regime.  For example, there is a lot of deciduous forest cover.  How much snow is there and when does it melt?  What are the key characteristics of regional climate?

P3, L18: Is this daily forcing data?  Please indicate in the ms.

P3, L25: this model is not using a snowmelt energy balance approach, and this needs to be rectified here and also justified.

P3, L30-31: The model is really using a grid, which is different than HRUs, so it should say "coarser grid". Also, what is "parameterization computation time"?

P3, L31 - P4, L1: There is a need for more detail on the model, what processes are represented, how it is run, etc., and then references can be added for further, more specific details.

P4, L4: What is the full reference for the "Natural Resources Canada" data?  This is needed.

P4, L11: despite analyzing 30-year average flows, the model simulations and the calibration approach uses daily flows, so the approach to neglect flow controls needs better justification.

Section 2.2: More details are needed on how the how the model was set up and parameterized, following the advice of reviewer #2. It isn't clear how parameters were set, how they varied among basins and HRUs, and how the HRUs were defined. In fact, the approach seems to be more consistent with a grouped response unit approach, where physical landscape groupings and their proportional area are derived from a grid, and parameters are set for the GRUs. Which parameters were important in the calibration and which were the results most sensitive to? Were there ranges that certain parameters were restricted to? Table 2 indicates that 17 parameters were determined by calibration alone, which provides a high potential for model equifinality, and so a more detailed explanation is important.

P4, L5: instead of each HRU, the percentages were determined for each grid cell.

P4, L27: Table 3 provides NSE and PBIAS info, not a set of model parameters.

P4, L28: the range is less than -15 to 15%, so why say this and not the actual range?

P4, L30-31: What can be said about how well the model represents processes within the watershed, such as snow accumulation and melt, for example? This relates back to the points about physical appropriateness of the model.

Section 2.3: The bias correction procedure is not adequately described. How well did the data compare and what type of correction was necessary? There is not enough information to determine what was done.

Section 2.4: This is still very unclear and there are no further details or reference provided to give more clarity. See comments by reviewer #2. A more clear illustration or some more detail may be needed to clearly explain this to the reader.

P5, L21: What is the Euclidean distance between pairs (i.e. in what space)?

P6, L5-8: Here is where internal climate variability is defined. So is this essentially the variation in forcing among the ensemble of climate model outputs?

P6, L11: what qualifies as a high flow? Is there a specific threshold?

P6, L22-27: These are important results and should be described in more detail. For instance, what are the magnitude and variability of the changes? How do the results differ among the ensemble members?

P6, L30: higher than what? Than the range of air temperature?

P7, L8: Are you referring to Jan-Feb streamflow? The section heading indicates that, but it isn't specified.

P7, L18: Instead of "majoritarily" it would be better to simply say "mostly" or "for the most part".

P8, L31-32: It is not clear why groundwater shows these differences, and this relates back to the need to explain how the models handles such processes.

P9, L7-8: monthly resolution of what? And what was the issue with representation of winter processes?

P9, L11-12: This is not correct. It is not clear how model structure and process representation affect the simulation of internal watershed processes, such as snowmelt and routing.

P9, L14-19: How confident can you be about the use of NRCANmet? Just because it is "widely used" isn't justification enough. There should be some indication somewhere about how well the simulations capture other variables (i.e. the internal watershed processes – especially snow accumulation and melt). Also, if measured Q is overestimated, would that not indicate that the problems with the model are even worse? Reviewer #1 raised some important concerns around the evaluation of the model.

P9, L27-31: This is unclear and could perhaps be better written.

P10, L23: high Z500 anomalies enhance what?

Section 4.3: It is not clear what the discussion here is getting at. Is it that variability among the ensemble members still predicts similar local weather patterns? Does this relate to the internal climate variability issue? It could be more clear.

P11, L2-3: Presumably, this is because only T and P data were fed into the model as forcing variables?

P11, L24: How do you know this correlation isn't an artifact of the model due to the representation of groundwater?

P11, L26-28: It was presumed that examining the influence of different weather patterns on streamflow regime was a purpose of the study. So what can be said about what the findings of this study suggest?

P11, L32: When it says "there will be less snow", it isn't clear initially that this isn't entirely a model projection, but that under the current climate the hydrological regime is less dominated by snowmelt runoff than the other basins. This could be rephrased to be more clear, although it becomes clear further into the paragraph.

P12, L23-28: The summary of results needs to be more clear. What do the "small" and "low" in parentheses mean? What are the percentages – just the number of runs that showed a certain category of streamflow change? This should be more clear. What is the threshold for categorizing large or small changes? And more importantly, what is the magnitude and variation of the projected changes? How confident are you in the various ensemble members, and are there some that are more likely than

others (i.e. these near the median) which should be given more weight or more consideration?  This is something to present more clearly in the results section, and convey briefly in the abstract.

P13, L2: what can be said about high flows from the results of this study?  What insight is there into changing flood regimes?

P13, L12: There is a need to indicate where the data can be found.  I think it is mostly publically available, so it should be a matter of identifying the sources.

Figure 1: It might be helpful to specify in the legend that the points are for the CRCM5-LE data.

Figure 6: is Delta Q annual?  It is not clear why it varies between 0 and 2.5 when in the next figure it varies between 0 and 1.  Also, to help, it could be made more clear by labelling the four weather classes in the right hand panel.

Figure 7: Instead of delta flow, this should be delta Q to be consistent with Fig. 6.

Figure 8: Units are needed throughout the figure and the legends.  Is part (a) surface temperature?

Figure 9: Units are needed throughout the figure and the legends.

Figure 10: The legend is ambiguous for the different classifications.  There are two red, three orange, two green and two blue categories.  Although the order seems to be clear, this could be improved.  Also, what are the vertical lines and hatches in the figure?  Standard deviations?  Please clarify.

Table 1: Where does urban landcover fit in?  How much is urban?

Table 2: The ranges in parameter values do not provide enough information.  What are the values for different HRUs (GRUs)?

Table 4: In the third column, what does the percentage in the top row specify?  % of the ensemble?  In the fourth to seventh columns, what are the units?  mm/day?  If so, what do the terms in brackets represent?  And why are some missing?

Grammatical and Technical Issues

P2, L8: "source" should be "sources"

P8, L26: instead of "expecting" it should say "projected"

P9, L13: "incriminated"?  Is there a better word choice?

P9, L16: "wrong measurements" should instead say "measurement uncertainty"

P9, L30: "which is conform"?  Pease rephrase.

P10,L1: "associated to stronger"; replace "to" with "with" and subsequently where the words "associated to" are used.

P10, L12: "enhance" should say "enhancement".

P10, L18: capital letter G for "great Lakes".

P11, L15: replace "in the meanwhile" with "Meanwhile"

P11, L19: either say "if more snow falls" or "if there is more snowfall"

P11, L26: "connexion" should be replaced with "connection" or "link"

P12, L17: the s should be removed from "precipitations", and where this is written subsequently.

---

## Author Response (AR2)

**We greatly appreciate input and comments by the reviewers and associate editor. In the new revised version, we have addressed almost all of them. Please find a point by point answer following by a marked up version of the revised manuscript:**

Dear authors,

Thank you for your responses to the reviewer comments and for the revision of the manuscript. Both reviewers were positive about the manuscript and recommended to publish it after some considerable revisions. The results showing increases in winter discharge in key watersheds within southern Ontario under future climate are of interest, and the study helps improve understanding of the potential hydrological impacts. However, the revised manuscript suffers from a number of issues of clarity and other problems, and will require further revisions to bring it up to the quality standards for this journal. Both reviewers offered constructive and helpful comments on how this paper needs to be improved, but I find these have not yet been fully addressed. A more careful and thorough effort is required.

Major Issues

1. The model must be more fully explained and justified as to its appropriateness for use in this region and under changing climates. Why is it appropriate for use here and what are its main limitations? Simply referring to the fact that others have used it and citing your earlier paper are not sufficient. The snowmelt routine appears to be a simple temperature index approach (i.e. a snow energy balance approach explicitly accounts for turbulent and radiative energy exchanges), and here the only inputs are temperature and precipitation. So why and how can this be justified under future climates?

**PRMS model has been used in this study because of its coupling capability with Modflow. The integrated model (GSFLOW) is planned to be used in this region in future studies (which would improve the results concerning groundwater flow). We added this justification to the manuscript. Temperature and precipitation are simple variables that have a direct impact on streamflow. A model with a greater number of variables will be more time consuming, especially multiplied by 50 members and it would lead to more uncertainties associated to each variable. Moreover, a recent study have shown that the snowmelt routine using temperature works well in the Big Creek watershed** (Champagne et al., 2019)**. These explanations were also added to the manuscript.**

2. There needs to be more detail and explanation of the model setup, parameterization, and calibration/validation. Both reviewers were adamant about this. More discussion is needed on model and data uncertainties, especially given the bias in simulated winter flows as noted by reviewer #1. The comments by reviewer #2 included a number of important issues to address

regarding model setup and geofabric. Why is it ok to neglect control structures and reservoirs, especially on the Grand River, where there are a number of flood control dams? (See specific comments further below.)

**Model setup and grid structure has been clarified and a figure has been added to the supplementary materials. We added more details of the model structure and the calibration/validation as suggested. Concerning the uncertainties associated to the dams the model has been calibrated and validated using the regulated flow series. Therefore, the dam effect if any should be implicitly accounted for during the model calibration. A more detailed study or analysis of control structure was not the main focus of this study. However, this is an important aspect and we will be exploring these aspects and impacts in a follow up future study**

3. The ascending hierarchical classification needs to be better described, and perhaps better illustrated, as it remains quite unclear. The reader needs to understand this. Why focus on runoff response groupings, given the non-linear nature of runoff, as opposed to strictly synoptic climatological patterns?

**We have renamed the ascending hierarchical classification as agglomerative hierarchical clustering, which correspond more adequately to the literature. The description of an AHC has been improved and a reference has been included. A runoff response grouping has been used to investigate if a similar change in streamflow can be associated to a similar change in atmospheric circulation. A group of runoff response can be associated to a mean streamflow change and a mean circulation change. It worked well with the extreme groups with clear atmospheric patterns for the largest and lowest increase in streamflow. This approach from the impact (streamflow) to the forcing (atmospheric circulation) may have more application than synoptic climatological patterns. It identifies groups of streamflow change than can be used independently of the atmospheric circulation. A sentence explaining this choice has been added to the manuscript.**

Detailed Comments

I refer to page and line numbers for the "clean" (i.e. non marked-up) version of the ms. My comments here are meant to help specifically address some of the reviewer concerns and to flag other issues that need to be dealt with.

P1, L12 and throughout: what is "internal variability of climate"? This term is central to the paper, but it isn't made entirely clear what this actual means. Does it refer to variation among the ensemble of climate model outputs?

**The internal variability of climate refers to the variability of climate due to inherently internal processes within the climate system. It is opposed to the variability of climate due to external forcing such as anthropogenic forcing (CO2 increase) or natural forcing (Volcanic eruptions and changes in solar radiation). A more clear definition of internal variability of climate has been added to the manuscript.**

P1, L18-22: the short summary of results is very unclear. What is meant by the terms in parentheses? The reader can't understand this by just reading the abstract. How significant is it that 14% of the ensemble members predict a high increase? What about the rest of the ensemble? More importantly, what is the magnitude and variation of projected flow changes?

**A clearer explanation of the results has been added to the abstract and especially the percentage change in streamflow for the different classes (and standard deviation).**

P1, L22: what does the 16% refer to? This isn't clear.

**This 16% corresponded to the number of members in the class HiQHiT (16% of the entire ensemble). This number has been removed to make the abstract clearer.**

P1, L22: what is "internal variability of hydrological projections"?

**How internal variability of climate will modulate hydrological projections is a more correct formulation. The sentence has been modified accordingly.**

P1, L30: the "choices" are really cascading sources of uncertainty throughout the modelling process. And does this link in to the concept of internal variability of climate? There is an opportunity to explain all of this more clearly.

**Internal variability of climate is one of the sources of uncertainty which cascades to the hydrological model simulations. How internal variability contributes to the total uncertainty has been explained more clearly.**

P2, L15: "future climate data should not be used…" – do you mean that coarse-scale climate model outputs shouldn't be used directly?

**We meant that the coarse scale climate model outputs shouldn't be used directly. This sentence has been modified to increase clarity.**

P2, L29: does a large ensemble assess the entire range of internal variability? Does this not also depend on selection of RCP, GCM, downscaling method, bias correction, hydrological model, parameterization, etc.?

**We agree that the total uncertainty depends on all of these factors but the uncertainties due to internal variability of climate (explained more clearly in the new revised of the manuscript) is a very specific type of uncertainty. It can be assessed by modifying the initial conditions of a GCM (model ensembles). The larger is the number of members in the ensemble, the higher is the range of internal variability that can be assessed. The ''entire'' range was perhaps an overstatement. It has been replaced with ''large'' in the revised manuscript.**

P2, L30: instead of "processes", do you mean "responses"?

**Responses is more appropriate here and has been used instead of processes.**

Introduction section: In general, this section should be a bit more clear on the overall purpose and objectives of this study, and on how it builds on previous work to advance understanding. What is new and how and why is it important?

**The objectives of the study section have been improved in the new revised version of the manuscript.**

P3, L8-9: What about other urban areas such as Kitchener–Waterloo and others?

**Other urban areas such as Guelph, Cambridge and Kitchener-waterloo were added here.**

Section 2.1: This section should describe the major landcover types in more detail, and a bit on the climate and the hydrological regime. For example, there is a lot of deciduous forest cover. How much snow is there and when does it melt? What are the key characteristics of regional climate?

**The description of the watersheds has been improved by including details about the amount of precipitation and snowfall per year, the spatial variability of precipitation, the type of flood regime as well as a more precise description of the land cover.**

P3, L18: Is this daily forcing data? Please indicate in the ms.

**It is daily forcing data. This information has been added.**

P3, L25: this model is not using a snowmelt energy balance approach, and this needs to be rectified here and also justified.

**This model uses the concepts of the energy balance approach but is using temperature as main driver of snow processes. This statement has been removed from the manuscript and more accurate description has been included.**

P3, L30-31: The model is really using a grid, which is different than HRUs, so it should say "coarser grid". Also, what is "parameterization computation time"?

**Term HRU has been replaced with grouped hydrological units (GRU) in the entire manuscript. The "Parametrization computation time" meant that Arcpy-GSFLOW has not been functional with a large number of GRU's. The sentence has been modified in the revised manuscript.**

P3, L31 - P4, L1: There is a need for more detail on the model, what processes are represented, how it is run, etc., and then references can be added for further, more specific details.

**A more detailed explanation of the model has been added to the manuscript as suggested.**

P4, L4: What is the full reference for the "Natural Resources Canada" data? This is needed.

**A full reference has been added to the manuscript.**

P4, L11: despite analyzing 30-year average flows, the model simulations and the calibration approach uses daily flows, so the approach to neglect flow controls needs better justification.

**The model has been calibrated and validated using the regulated flow series. Therefore, the dam effect, if any, should be implicitly accounted for during the model calibration. We assume that the flow control will not be modified in the future and that the relative change in streamflow will not be impacted by the dams.**

Section 2.2: More details are needed on how the how the model was set up and parameterized, following the advice of reviewer #2. It isn't clear how parameters were set, how they varied among basins and HRUs, and how the HRUs were defined. In fact, the approach seems to be more consistent with a grouped response unit approach, where physical landscape groupings and their proportional area are derived from a grid, and parameters are set for the GRUs. Which parameters were important in the calibration and

which were the results most sensitive to? Were there ranges that certain parameters were restricted to? Table 2 indicates that 17 parameters were determined by calibration alone, which provides a high potential for model equifinality, and so a more detailed explanation is important.

**The word HRU has been replaced by GRU in the entire manuscript. More details of model setup and parametrization have been included. The spatial variability of the parameters has been added in the supplementary materials. A sensitivity analysis of the parameters has been performed for the Big Creek watershed and results has been added in the supplementary material as well.**

P4, L5: instead of each HRU, the percentages were determined for each grid cell.

**HRU has been replaced by GRU to show that grid cells were used.**

P4, L27: Table 3 provides NSE and PBIAS info, not a set of model parameters.

**This was a mistake. We were referring to Table 2. This has been modified in the manuscript.**

P4, L28: the range is less than -15 to 15%, so why say this and not the actual range?

**The reference to -15 to 15% was used because it is considered as a good fit according to Moriasi et al. (2007).**

P4, L30-31: What can be said about how well the model represents processes within the watershed, such as snow accumulation and melt, for example? This relates back to the points about physical appropriateness of the model.

**More details on the calculations of the snow processes have been added to the manuscript. The representation of snow has been tested in Big Creek watershed** (Champagne et al., 2019) **and was satisfactory. A reference to this study has been added to the manuscript.**

Section 2.3: The bias correction procedure is not adequately described. How well did the data compare and what type of correction was necessary? There is not enough information to determine what was done.

**The description of the bias correction technique has been improved and the comparison between bias corrected and raw data has been added to the supplementary materials.**

Section 2.4: This is still very unclear and there are no further details or reference provided to give more clarity. See comments by reviewer #2. A more clear illustration or some more detail may be needed to clearly explain this to the reader.

**This section has been rewritten for clarity. The ascending hierarchical classification has been renamed agglomerative hierarchical clustering which correspond to more references in the literature. A reference has been also added.**

P5, L21: What is the Euclidean distance between pairs (i.e. in what space)?

**The intraclass variance has been used at each grouping step. The term Euclidean distance has been removed from the manuscript**

P6, L5-8: Here is where internal climate variability is defined. So is this essentially the variation in forcing among the ensemble of climate model outputs?

**The internal variability of climate is the variability of climate not due to forcing but only due to the chaotic variability of atmospheric circulation. The explanation of internal variability has been improved through the manuscript. It has now been explained earlier in the text.**

P6, L11: what qualifies as a high flow? Is there a specific threshold?

**High flows are defined as streamflow higher than a threshold corresponding to the average streamflow plus three times the standard deviation using the observation streamflow. The description of the high flows has been added to the revised manuscript.**

P6, L22-27: These are important results and should be described in more detail. For instance, what are the magnitude and variability of the changes? How do the results differ among the ensemble members?

**The 50-members average changes and standard deviation has been added to the manuscript. This gives information on how the results differ between classes and inside classes.**

P6, L30: higher than what? Than the range of air temperature?

**Change in precipitation appear more variable between members compare to the change in temperature. This has been described in the revised manuscript**

P7, L8: Are you referring to Jan-Feb streamflow? The section heading indicates that, but it isn't specified.

**Yes we are referring to January-February. January-February streamflow has been specified in the manuscript**

P7, L18: Instead of "majoritarily" it would be better to simply say "mostly" or "for the most part".

**We replaced majoritarily by the "most part" in the manuscript as suggested.**

P8, L31-32: It is not clear why groundwater shows these differences, and this relates back to the need to explain how the models handles such processes.

**Big Creek shows a lower increase in overall streamflow. Therefore, it is likely that change in groundwater flow is also lower. This has been added to the manuscript.**

P9, L7-8: monthly resolution of what? And what was the issue with representation of winter processes?

**It was mostly lack of ponding and frozen soil in the HydroGeosphere model that may have overestimated streamflow in winter. According to** Erler et al., (2018) **frozen soil may delay the streamflow due to more ponding. We have mentioned these aspects in the revised manuscript.**

P9, L11-12: This is not correct. It is not clear how model structure and process representation affect the simulation of internal watershed processes, such as snowmelt and routing.

**Not all processes can be compared to observations but Snowpack was satisfactorily simulated by PRMS model using NRCANmet in Big Creek watershed. This statement has been added to the manuscript.**

P9, L14-19: How confident can you be about the use of NRCANmet? Just because it is "widely used" isn't justification enough. There should be some indication somewhere about how well the simulations capture other variables (i.e. the internal watershed processes – especially snow accumulation and melt). Also, if measured Q is overestimated, would that not indicate that the problems with the model are even worse? Reviewer #1 raised some important concerns around the evaluation of the model.

**Snow accumulation is adequately simulated by PRMS model when forced by NRCANmet as its have been shown by** Champagne et al. (2019)**. Measured Q may be overestimated during ice conditions but the model was calibrated using these possibly overestimated values. The actual**

**discharge, taking into consideration ice on the river, could be used to calibrate the model, but this is not likely to improve the ability of the model to simulate winter streamflow. The statement on stream ice is aimed to show that measurements can also have errors or uncertainties but not to explain why the model overestimates streamflow.**

P9, L27-31: This is unclear and could perhaps be better written.

**This part has been rewritten.**

P10, L23: high Z500 anomalies enhance what?

**The Z500 anomalies are increasing. It has been reformulated to increase the clarity.**

Section 4.3: It is not clear what the discussion here is getting at. Is it that variability among the ensemble members still predicts similar local weather patterns? Does this relate to the internal climate variability issue? It could be more clear.

**The goal of this paragraph is to discuss the method of classes used in this study. There is a large atmospheric variability between members of the same class that produce similar local conditions. Some sentences were rewritten to increase clarity.**

P11, L2-3: Presumably, this is because only T and P data were fed into the model as forcing variables?

**The goal of this paragraph is to discuss the role of temperature and precipitation that directly impact streamflow in the same months opposed to delays in the relationships due to snow process and groundwater. The mention of January-February has been added to increase the clarity.**

P11, L24: How do you know this correlation isn't an artifact of the model due to the representation of groundwater?

**PRMS model is not adapted to answer this question. A surface-groundwater coupled model such as GSFLOW is therefore suggested to be used to confirm this hypothesis.**

P11, L26-28: It was presumed that examining the influence of different weather patterns on streamflow regime was a purpose of the study. So what can be said about what the findings of this study suggest?

**This study didn't examine the atmospheric conditions that occurred in the previous months (November-December) while they could have an impact in the modulation of streamflow in January-February. The results in the lags between precipitation and January- February suggest that the succession of different atmospheric patterns in the previous months can have an impact on the January-February modulation of streamflow.**

P11, L32: When it says "there will be less snow", it isn't clear initially that this isn't entirely a model projection, but that under the current climate the hydrological regime is less dominated by snowmelt runoff than the other basins. This could be rephrased to be more clear, although it becomes clear further into the paragraph.

**This sentence has been reformulated to improve clarity.**

P12, L23-28: The summary of results needs to be more clear. What do the "small" and "low" in parentheses mean? What are the percentages – just the number of runs that showed a certain category of streamflow change? This should be more clear. What is the threshold for categorizing large or small changes? And more importantly, what is the magnitude and variation of the projected changes? How confident are you in the various ensemble members, and are there some that are more likely than others (i.e. these near the median) which should be given more weight or more consideration? This is something to present more clearly in the results section, and convey briefly in the abstract.

**The summary of results has been greatly improved and suggested aspects have been included.**

P13, L2: what can be said about high flows from the results of this study? What insight is there into changing flood regimes?

**This study did not focus on high flows. We can hypothesize that average change in atmospheric circulation and associated local temperature/precipitation in winter will likely produce more high flows. However, as stated in the conclusion, the day to day variability of atmospheric circulation needs to be studied for an estimation of high flows variability. This has been clarified in the new version of the manuscript.**

P13, L12: There is a need to indicate where the data can be found. I think it is mostly publically available, so it should be a matter of identifying the sources.

**The data sources were added.**

Figure 1: It might be helpful to specify in the legend that the points are for the CRCM5-LE data.

**The mention of CRCM5-LE was added to the legend.**

Figure 6: is Delta Q annual? It is not clear why it varies between 0 and 2.5 when in the next figure it varies between 0 and 1. Also, to help, it could be made more clear by labelling the four weather classes in the right hand panel.

**It varies between -2.5 and 2.5 because it represented the normalized change in streamflow used to classify the members, not the absolute change in streamflow. Normalized data has been replaced by absolute values in the new figure and the four weather classes have been labelled.**

Figure 7: Instead of delta flow, this should be delta Q to be consistent with Fig. 6.

**Delta flow has been replaced by delta Q in the figure.**

Figure 8: Units are needed throughout the figure and the legends. Is part (a) surface temperature?

**The units were added to the figure and the word ''surface'' has been added to temperature.**

Figure 9: Units are needed throughout the figure and the legends.

**The units have been added to the new version of the manuscript**

Figure 10: The legend is ambiguous for the different classifications. There are two red, three orange, two green and two blue categories. Although the order seems to be clear, this could be improved. Also, what are the vertical lines and hatches in the figure? Standard deviations? Please clarify.

**Different line types have been included for the streamflow classes to improve the clarity. The hatches are the standard deviation for each class. It has been included in the legend.**

Table 1: Where does urban landcover fit in? How much is urban?

**Urban is counted as barren. Most of barrens are actually urban. This information has been added to the table.**

Table 2: The ranges in parameter values do not provide enough information. What are the values for different HRUs (GRUs)?

**The average for each parameter has been added as it gives more information. However, the values for the different GRUs cannot be given as there is thousands of GRUs in this watershed. They are shown in maps in the supplementary materials.**

Table 4: In the third column, what does the percentage in the top row specify? % of the ensemble? In the fourth to seventh columns, what are the units? mm/day? If so, what do the terms in brackets represent? And why are some missing?

**The third column indicates the percentage of the ensemble. This information has been added to the table. The unit was mm/day but has been replaced by percentage of increase as it represents better the relative change of streamflow. The term in parenthesis is the standard deviation. This information has been added in the table. Standard deviation are not available for classes that include only one member.**

Grammatical and Technical Issues

P2, L8: "source" should be "sources"

P8, L26: instead of "expecting" it should say "projected"

**Changes were done as suggested**

P9, L13: "incriminated"? Is there a better word choice?

**Incriminated has been changed to "was a source of error"**

P9, L16: "wrong measurements" should instead say "measurement uncertainty"

**Changed as suggested**

P9, L30: "which is conform"? Pease rephrase.

**This sentence has been reformulated**

P10,L1: "associated to stronger"; replace "to" with "with" and subsequently where the words "associated to" are used.

**Associated to has been replaced by associated with in the entire manuscript**

P10, L12: "enhance" should say "enhancement".

P10, L18: capital letter G for "great Lakes".

P11, L15: replace "in the meanwhile" with "Meanwhile"

P11, L19: either say "if more snow falls" or "if there is more snowfall"

P11, L26: "connexion" should be replaced with "connection" or "link"

P12, L17: the s should be removed from "precipitations", and where this is written subsequently.

**All these changes have been done in the manuscript**

[revised manuscript text omitted]
) in southern Ontario were selected for this study considering for their long hydrometric data time series archives and representation of well the diversity of spatial scales, soil type, and land use in this region (Figure 1 and Table 1). Agriculture activity is the largest land use category in all four watersheds, covering more. than 80% of the entire surface in Big Creek, Thames River and Grand River. Credit River has the highest proportion of forest (32%), mostly deciduous species. TwoSeveral major urban areascities , are located in the study

25 area: Brantford, Cambridge, Kitchener-Waterloo and Guelph in the along the Grand River watershed;, and and London in along the Thames River. are located in the study area and aAdditional urban areas are located in the Credit river watershed in the vicinity of the Greatest Toronto Area while . Tthe Big Creek watershed contains the lowest proportion of urbanization (2%). These watersheds also vary in soil type: sand predominates in Big Creek (79%) and Credit River (43%), but a large area of Credit River is also covered by loamy soil (49%). Grand River has almost an equal proportion of sand (30%), loam (32%)

30 and clay (38%), while Thames River contains more clay (39%). The elevation is also highly variable with the highest altitudes in the North parts of Grand River (531 m) and Credit River (521 m) while the lowest areas are located in the sandplains further south in Grand River (178 m) and Big Creek (179 m).

**2.2 PRMS hydrological model**

The Precipitation Runoff Modelling System (PRMS), a semi-distributed conceptual hydrological model developed by Leavesley et al. (1983), was applied in all four watersheds to simulate the future evolution of streamflow for each member of a large climate ensemble. PRMS is used in this study because it needs only basic daily forcing climate data (minimum and maximum temperature, and precipitation). The advantage of using a model that need only few data as input is that it reduces uncertainties from multiple variables and reduce the model computational time. A drawback of using temperature is that energy balance is not physically represented. However, in an earlier study in the Big Creek watershed, PRMS represented well the snow processes (Champagne et al., 2019) showing that the use of temperature and precipitation are satisfactory to represent the snow processes in this region. Moreover, PRMS can be coupled with MODFLOW groundwater model (GSFLOW) to study the interaction between surface and groundwater flow (Markstrom and Regan, 2008). While MODFLOW was not activated in this study, having PRMS set up in these watersheds will facilitate the use of GSFLOW in future studies. This model , and has been widely applied in watersheds that experience are affected by periodic snowfall (Dressler et al., 2006; Liao and Zhuang, 2017; Mastin et al., 2011; Surfleet et al., 2012; Teng et al., 2017, 2018). PRMS was used to and study in an earlier study The hydrological calculations in PRMS are based on physical laws and empirical relations between measured and estimated quantities. A series of hydrologic reservoirs are used (plant canopy interception, snowpack, soil zone, subsurface) are used in the model and the water flowing between the reservoirs are computed for each grouped hydrological response units (GHRUs). In this study the potential evapotranspiration was estimated using the Jensen-Haise formulation (Jensen and Haise, 1963). The interception was calculated separately for summer rain, winter rain and winter snow and was a function of the plant type. The separation between rainfall and snowfall was done by the snow module using temperature thresholds. If a day has a maximum temperature below 0 °C, all precipitation of the day was considered as snow. If a day has a minimum temperature higher than 0 °C and a maximum temperature higher than a threshold to calibrate, then all precipitation is considered rain. A mixed precipitation is computed when conditions are between these values. The snowpack dynamics are simulated through estimate of energy and water dynamics. The energy available to melt the snow is based on estimation of shortwave radiation, longwave radiation, convection and condensation. Shortwave solar radiation was estimated using a degree-day method. Longwave radiation is the integration of the longwave radiation from the land cover and from the air depending on the emissivity of air. Convection and condensation are computed together as a function of temperature and a calibrated coefficient. Surface runoff due to infiltration excess (Hortonian runoff) is computed using the antecedent soil moisture content. The amount of water not contributing to Hortonian runoff is infiltrated and directed to the soil zone. The soil zone module computes transpiration, recharge to the groundwater reservoir and three components of the streamflow: saturation excess (Dunnian runoff), subsurface flow through soil cracks, animal borrows or leaf litter (fast interflow) and subsurface flow (slow interflow). These processes are described in more details by Markstrom et al. (2015). For more information about the structure of a recent version of PRMS, reader is referred to Markstrom et al., (2015). A major advantage of this model used in a climate change impact study is the representation of snowmelt using an energy balance approach based on temperature and

[revised manuscript text omitted]
. In a general concept, tThe AHC calculates first the Euclidean distancevariance between each pair of observationsmembers. The pair with the lowest closest Euclidean distance is variance  mergesd into a single class. The Euclidean distance of this class is then calculated by averaging the Euclidean distance between each member of this class and all other members. In the next step, tThe next pair of classes or pair of observations members with the smallest Euclidean distance is merged and averaged similarly that would result in the smallest increase of total variance, compared to the previous step, is grouped together. This process is repeated 49 times, until all classes of members have been merged into a single class. The classification was used to simplify the study of the connections between the future change in large scale atmospheric circulation, local meteorological conditions and streamflow. In this study, tThe AHC was applied first to the all four 4 watershed's s January-February normalized change in of streamflow and then to the 4 four watershed'ss 
[revised manuscript text omitted]
 | 7,9,10,18, 29,35,40 | 14% | 8.3 (7.8) | 18.8 (5.8) | 17.8 (6.4) | 18.6 (6.5) |

---

## Author Response (AR3)

**We would like to thank again the editor and reviewers for their constructive comments on our paper. Please find below the answer to the editor comments, following by an annotated version of the manuscript:**

5 Comments to the Author:

Dear authors,

Thank you for your work to revise the paper. It has been greatly improved and is now ready for final publication. I have some final technical comments:

1. In the intro, you say "Great Lakes region contains ~20 per cent of the world's freshwater", which is not correct.
10 (Maybe you mean % of surface freshwater apart from glaciers and ice sheets, or something else?)

**" ~20 per cent of the world's freshwater" has been replaced by "~20 per cent of the world's unfrozen surface freshwater"**

2. In section 2.1 you mention "Greatest Toronto Area"; this should be "Greater Toronto Area".

**The change has been done as suggested**

[revised manuscript text omitted]
** | 7,9,10,18, 29,35,40 | 14% | 8.3 (7.8) | 18.8 (5.8) | 17.8 (6.4) | 18.6 (6.5) |